# Learning to Insert for Constructive Neural Vehicle Routing Solver

**Fu Luo**[1,2]**, Xi Lin**[3]**, Mengyuan Zhong**[4]**, Fei Liu**[3]**, Zhenkun Wang**[1,2*]**, Jianyong Sun**[4*]**, Qingfu Zhang**[3]

[1] School of Automation and Intelligent Manufacturing,
Southern University of Science and Technology, Shenzhen, China
[2] Guangdong Provincial Key Laboratory of Fully Actuated System Control Theory and Technology,
Southern University of Science and Technology, Shenzhen, China
[3] Department of Computer Science, City University of Hong Kong, Hong Kong SAR, China
[4] School of Mathematics and Statistics, Xi'an Jiaotong University, Xi'an, China

`luof2023@mail.sustech.edu.cn, xi.lin@my.cityu.edu.hk, my.zhong@stu.xjtu.edu.cn,`
`fliu36-c@my.cityu.edu.hk, wangzhenkun90@gmail.com,`
`jy.sun@xjtu.edu.cn, qingfu.zhang@cityu.edu.hk`

## Abstract

Neural Combinatorial Optimisation (NCO) is a promising learning-based approach for solving Vehicle Routing Problems (VRPs) without extensive manual design. While existing constructive NCO methods typically follow an appending-based paradigm that sequentially adds unvisited nodes to partial solutions, this rigid approach often leads to suboptimal results. To overcome this limitation, we explore the idea of the insertion-based paradigm and propose Learning to Construct with Insertion-based Paradigm (L2C-Insert), a novel learning-based method for constructive NCO. Unlike traditional approaches, L2C-Insert builds solutions by strategically inserting unvisited nodes at any valid position in the current partial solution, which can significantly enhance the flexibility and solution quality. The proposed framework introduces three key components: a novel model architecture for precise insertion position prediction, an efficient training scheme for model optimization, and an advanced inference technique that fully exploits the insertion paradigm's flexibility. Extensive experiments on both synthetic and real-world instances of the Travelling Salesman Problem (TSP) and Capacitated Vehicle Routing Problem (CVRP) demonstrate that L2C-Insert consistently achieves superior performance across various problem sizes. The code is available at https://github.com/CIAM-Group/L2C_Insert.

## 1 Introduction

The vehicle routing problem (VRP) [1] is an important combinatorial optimization problem with extensive real-world applications in domains such as transportation [2], logistics [3], and circuit design [4]. Due to its NP-hard nature [5], obtaining exact VRP solutions is computationally difficult [5]. The existing approaches, therefore, focus on finding approximate solutions within a practical runtime using problem-specific heuristics. However, designing these heuristics requires substantial domain expertise. Recently, Neural Combinatorial Optimization (NCO) methods have emerged as a promising alternative, as they can leverage neural networks to automatically learn effective heuristics from data, thereby significantly reducing the need for extensive manual design [6–18].

Among NCO methods, constructive approaches attract considerable research interest [6, 19–26]. These methods typically employ an appending-based constructive paradigm that sequentially selects

---
*Corresponding author

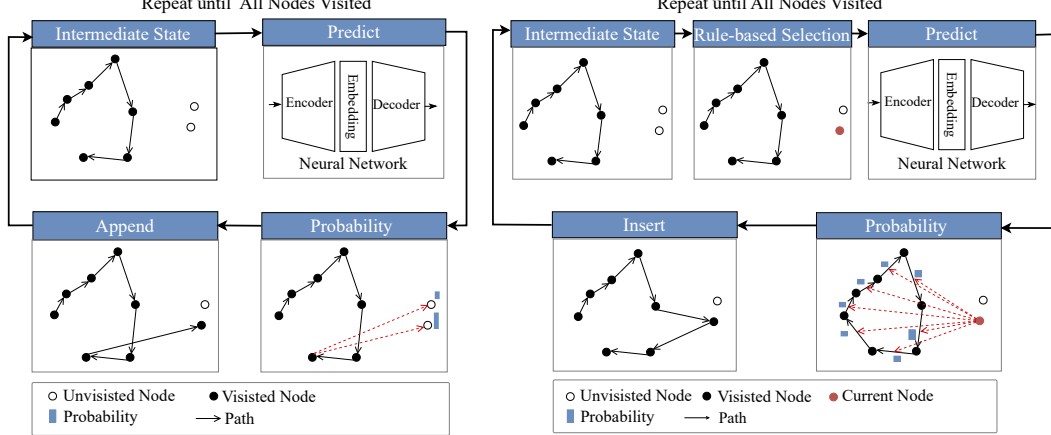

(a) Appending-based Construction for NCO (**Conventional**)      (b) Insertion-based Construction for NCO (**Unexplored**)

Figure 1: **Appending-based construction vs. Insertion-based construction.** Given the same intermediate state in the construction process: **(a)** The appending-based paradigm restricts the addition of a selected node exclusively to the end of the current partial solution. This inflexibility leads to path intersections, a common cause of suboptimal solutions. **(b)** In contrast, the insertion-based construction offers greater flexibility by allowing the selected node to be inserted into any feasible position within the partial solution. Such flexibility is crucial for avoiding the intersections and thereby enhancing the potential for generating high-quality solutions.

unvisited nodes and appends them to the end of the current partial solution. This paradigm is inspired by the sequence-to-sequence framework [27] in natural language processing (NLP), where the model encodes an input sequence into an embedding and then autoregressively generates an output sequence based on this representation. Vinyals et al. [6] pioneer this adaptation through their Pointer Networks (Ptr-Nets), which establishes the first sequence-to-sequence framework specifically for neural combinatorial optimization. Subsequent research has further advanced constructive NCO methods within this appending-based paradigm, with improvements focusing on model training techniques [7] and architectural enhancements [19]. However, appending-based methods suffer from a fundamental limitation due to their strict operational constraint that only permits adding new nodes to the end of the current partial solution. This constraint prevents potentially more effective modifications and often leads to suboptimal partial solutions caused by early greedy decisions.

The insertion-based paradigm offers a more effective alternative for addressing VRPs [28, 29]. This approach constructs solutions by sequentially inserting an unvisited node into any valid position within the current partial solution, which potentially offers greater flexibility and yields higher-quality solutions than the appending-based paradigm. For example, as shown in Figure 1, given the same intermediate state in the construction process, the rigid sequential nature of the appending paradigm can only add the selected node to the end of the solution, leading to path intersections and a suboptimal solution. In contrast, the insertion-based construction allows for the flexible placement of the selected node into any feasible position within the partial solution path, thereby avoiding such intersections and holding the potential to generate higher-quality solutions. Despite these clearly demonstrated benefits, the insertion-based constructive approach has remained unexplored in NCO for VRPs in the past decade. A comprehensive review of constructive NCO methods can be found in Appendix A.

To bridge this research gap, this work proposes Learning to Construct with Insertion-based Paradigm (L2C-Insert), a novel insertion-based learning framework for constructive NCO. To the best of our knowledge, L2C-Insert represents the first learning-driven, insertion-based framework specifically designed for solution construction in NCO. Our main contributions can be summarized as follows:

- We propose a novel model architecture, an effective training scheme, and a powerful inference technique for the L2C-Insert framework. Together, these components are crucial for enabling the L2C-Insert framework to achieve robust performance.

- We develop a supervised learning based training scheme tailored for the efficient learning of the L2C-Insert model. Building on the inherent flexibility of the insertion paradigm,

we further propose an insertion-based local reconstruction mechanism to enhance model performance during inference.

- We conduct comprehensive experiments to explore the advantages of the L2C-Insert framework on both synthetic and real-world benchmarks. The results demonstrate that our L2C-Insert method achieves outstanding performance on TSP and CVRP instances with sizes ranging from 100 to 100K nodes.

## 2 Preliminaries

### 2.1 VRP Formulation

A VRP instance $S$ can be represented as a graph containing $n$ nodes, where each node $i \in \{1, \ldots, n\}$ is represented by a feature vector $\boldsymbol{s}_i \in \mathbb{R}^{d_s}$. For example, in the TSP, $\boldsymbol{s}_i$ typically corresponds to the 2D coordinates of node $i$. A solution to a VRP instance specifies the sequence, or set of sequences (routes/tours), in which nodes are visited. For example, in the TSP case, a solution is a single tour that can be represented as a permutation of all $n$ nodes: $\boldsymbol{\pi} = (\pi_1, \ldots, \pi_t, \ldots, \pi_n)$, where each element $\pi_t \in \{1, \ldots, n\}$ is a distinct node. The objective in solving VRPs is generally to minimize the cost of the solution, which often corresponds to the total Euclidean length of the tour(s). A VRP solution is considered feasible if it satisfies all problem-specific constraints. For example, in the TSP, a feasible solution must visit each node in the graph exactly once. Constraints in the CVRP additionally include vehicle capacity limits, which are further detailed in Appendix B.

### 2.2 Appending-based Construction for Solving VRPs

The appending-based constructive heuristic is a prominent approach that NCO methods frequently leverage for solving VRPs [6, 19, 21–23, 30–33]. Initializing with an empty solution, this heuristic constructs the solution to a given instance by sequentially appending unvisited nodes to the end of the current partial solution until all nodes are visited. At each construction step, it selects a node from the set of unvisited nodes to append, based on a specified strategy. A common selection strategy is to choose the unvisited node that has the smallest Euclidean distance to the last node in the current partial solution. However, this simple selection strategy typically leads to solutions with relatively low quality. The existing learning-based constructive NCO methods focus on using neural networks to learn more effective selection strategies capable of generating near-optimal solutions.

### 2.3 Insertion-based Construction for Solving VRPs

The insertion-based constructive heuristic, though well-established in Operations Research for solving VRPs [34, 35, 29], has not been explored for constructive NCO. Similar to the appending-based approach, this heuristic begins with an empty solution and iteratively incorporates unvisited nodes into the partial route until all nodes are included. Unlike simple appending, this method permits insertion of unvisited nodes at any valid position within the current partial solution. Each construction step involves two critical decisions: first, selecting an unvisited node for insertion, and second, identifying the proper position within the current partial solution for insertion. Typical node selection strategies include random selection from the unvisited set or choosing the node closest to the previously inserted node. Regarding position selection, a common heuristic is to choose the insertion node that minimizes the increase in the total solution cost (e.g., tour length) upon inserting the selected node. Our work mainly focuses on leveraging neural networks to learn more effective position selection strategies to generate high-quality solutions.

## 3 L2C-Insert: Learning to Construct in Insertion Paradigm

This section proposes the L2C-Insert framework for learning the insertion-based construction heuristic, which integrates three key components: (1) a novel model architecture, (2) an effective training scheme, and (3) a powerful inference technique. We demonstrate this framework through solving TSP, and the required adaptations for solving CVRP are provided in Appendix B.

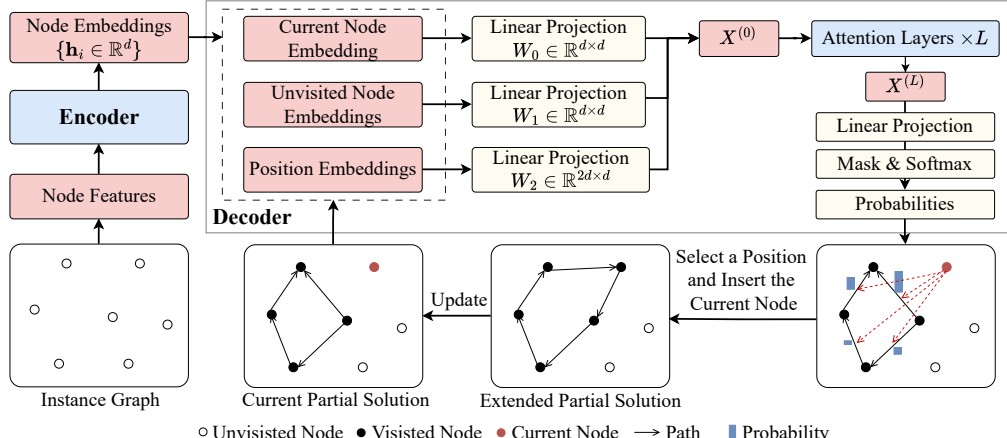

Figure 2: **Model Structure of L2C-Insert. (a) Encoding:** At the beginning of solution construction, the encoder transforms each node's feature vector into a $d$-dimensional node embedding $\mathbf{h}_i \in \mathbb{R}^d$. **(b) Decoding:** At construction step $t$, the decoder maintains three embeddings for 1) the current node (the node to be inserted), 2) unvisited nodes, and 3) the positions in the partial solution, where each position embedding is created by horizontally concatenating the embeddings of adjacent visited nodes. Each embedding type undergoes separate linear transformations before being collectively processed through $L$ stacked attention layers. The final insertion probabilities are obtained by applying a linear projection followed by a masked softmax operation to the refined embeddings.

## 3.1 Model Structure

Similar to the appending-based constructive NCO models, our L2C-Insert model utilizes an encoder-decoder framework, with the decoder features with multiple attention layers [36] for the promising learning ability [22, 23] as shown in Figure 2.

**Encoder** The encoder contains a linear projection layer followed by an attention layer. For a problem instance $S$ with $n$ nodes, each node's feature vector $\boldsymbol{s}_i \in \mathbb{R}^{d_s}$ (where $i = 1, \ldots, n$) is first transformed into an initial node embedding $\mathbf{h}_i^0 \in \mathbb{R}^d$, where $d$ denotes the embedding dimension. These initial embeddings are subsequently updated through the attention layer [21, 22] to produce the refined embeddings $\mathbf{h}_i \in \mathbb{R}^d$, which is the output of the encoder.

**Decoder** At construction step $t$, the decoder receives the embeddings of all nodes, denoted as $\mathbf{h}_i \in \mathbb{R}^d, i = 1, \ldots, n$, the current partial solution $(\pi_1, \pi_2, \ldots, \pi_{t-1})$, and the set of unvisited nodes $\{u_j, u_j \notin \{\pi_{1:t-1}\}\}$. To compute the probability of insertion at the current step, embeddings for three distinct inputs are required: 1) the current node (the node to be inserted), 2) unvisited nodes, and 3) the positions in the partial solution.

First of all, the unvisited node with the minimum Euclidean distance to the previously inserted node is selected as the current node to be inserted, of which the embedding is denoted as $\mathbf{h}_{u_c}$. Secondly, the embeddings of the remaining unvisited nodes can be denoted as $H_u = \{\mathbf{h}_{u_j}\}, u_j \notin \{\pi_{1:t-1}\}, u_j \neq u_c$. Two learnable linear projections $W_0 \in \mathbb{R}^{d \times d}$ and $W_1 \in \mathbb{R}^{d \times d}$ are applied to the current node's embedding and each unvisited node's embedding, respectively. Thirdly, to generate the embedding of a potential insertion position within the partial solution, the embeddings of adjacent visited nodes are horizontally concatenated. Given the set of visited node embeddings $\{\mathbf{h}_{\pi_i}\}$, the embedding for the insertion position in the solution segment $(\pi_i, \pi_{i+1})$ is computed as:

$$\mathbf{e}_{\pi_i} = W_2 \begin{cases} [\mathbf{h}_{\pi_i}, \mathbf{h}_{\pi_{i+1}}] & i \in \{1 : t-2\}, \\ [\mathbf{h}_{\pi_{t-1}}, \mathbf{h}_{\pi_1}], & i = t-1, \end{cases} \tag{1}$$

where $[\cdot, \cdot]$ represents the horizontal concatenation operator. This operator produces an initial $2d$-dimensional initial insertion position embedding, which is subsequently projected to a $d$-dimensional vector via the trainable matrix $W_2 \in \mathbb{R}^{2d \times d}$ to maintain dimensional consistency with node embeddings. The set of position embeddings is denoted as $H_e = \{\mathbf{e}_{\pi_i}\}, i \in \{1 : t-1\}$.

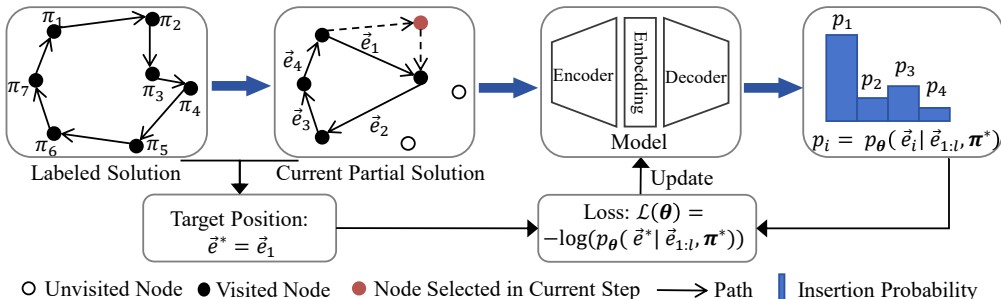

Figure 3: **Training Scheme for L2C-Insert Model.** The model learns to sequentially insert the current node into the target position in the partial solution. The target position for the current node is determined by its adjacent visited nodes in the original labeled solution. The training objective is to maximize the probability of the model predicting the correct target insertion position, as defined by the provided loss function.

Then, these embeddings together are refined by $L$ stacked attention layers. The first layer receives input $X^{(0)} = \{\mathbf{h}_c, H_e, H_u\}$, with each subsequent $j$-th attention layer processing the output from the $(j-1)$-th layer, culminating in the final output $X^{(L)}$. Then, a linear projection and softmax function are applied to $X^{(L)}$ to compute the insertion probabilities of the current node into all valid positions. The masking operation assigns $-\infty$ to invalid positions (those corresponding to the current node and non-visited nodes) to ensure proper probability normalization over valid insertion locations. The inserting probability $p_i$ is calculated as:

$$
\begin{aligned}
x_i &= X_i^{(L)} W_f + b_f, \\
a_i &= \begin{cases} x_i & \forall i \in \{\pi_{1:t-1}\}, \\ -\infty & \text{otherwise}, \end{cases} \\
\mathbf{p} &= \text{Softmax}(\mathbf{a}),
\end{aligned} \tag{2}
$$

where $W_f \in \mathbb{R}^{d \times 1}$ and $b_f \in \mathbb{R}^1$ are learnable parameters. Each $p_i \in \mathbf{p}$ denotes the probability of inserting the current node between consecutive nodes $(\pi_i, \pi_{i+1})$ in the partial solution. The decoder executes this sampling-and-insertion procedure $n$ times to construct the complete solution $\{\pi_1, \cdots, \pi_n\}$.

## 3.2 Training Scheme

Constructive NCO models typically adopt either reinforcement learning (RL) or supervised learning (SL) methods for training. The SL-based method can effectively train the model with a heavy decoder, whereas the RL-based method is not capable, as indicated by Luo et al. [22]. Since our model utilizes an insertion-based construction process, which differs significantly from the traditional appending-based approaches, we redefine the SL-based training scheme as follows.

For a problem instance with $n$ nodes and its labeled solution $\boldsymbol{\pi}^*$ (e.g., an optimal tour), we first randomly select a node as the initial partial solution while designating the remaining nodes as unvisited. Guided by the original labeled solution, the model then learns to sequentially re-insert these unvisited nodes into their target positions within the evolving partial solution. The target position for an unvisited node is defined by the two visited nodes in the current partial solution that are adjacent to this unvisited node in the complete labeled solution. For example, consider the partial solution illustrated in Figure 3, where $\pi_2$ represents the current node. After excluding the unvisited nodes $\pi_4$ and $\pi_5$, the two visited nodes adjacent to $\pi_2$ in the labeled solution are $\pi_1$ and $\pi_3$. Therefore, the target position is $\vec{e}^* = \vec{e}_1 = (\pi_1, \pi_3)$. The loss function is then computed as:

$$
\mathcal{L}(\boldsymbol{\theta}) = -\log p_{\boldsymbol{\theta}} \left( \vec{e}^* \mid \vec{e}_{\pi_{1:l}}, \boldsymbol{\pi}^* \right), \tag{3}
$$

where $\{\vec{e}_{\pi_{1:l}}\}$ is the set of positions in the current incomplete solution, $\vec{e}^* \in \{\vec{e}_{\pi_{1:l}}\}$ is the target position, and $p_{\boldsymbol{\theta}} \left( \vec{e}^* \mid \vec{e}_{\pi_{1:l}}, S \right)$ is the probability of inserting the current node into target psition predicted by the model parameterized by $\boldsymbol{\theta}$. Subsequently, the model parameters are optimized using a gradient-based algorithm (e.g., Adam [37]) to minimize the loss function. Through this

training process, the model progressively learns to accurately position previously unvisited nodes into their correct positions within the solution sequence, ultimately acquiring the capability to generate high-quality solutions.

## 3.3 Insertion-based Local Reconstruction

For NCO, local reconstruction techniques are widely employed during the inference phase to improve solution quality through iterative refinement of local solution segments [38, 39, 22, 40]. While these methods can effectively trade computational time for enhanced solution quality, they face inherent limitations when applied to appending-based constructive models. Specifically, the constraint of using sequence-based destruction strategies severely limits both the destruction space and the subsequent reconstruction action space, often trapping solutions in local optima. For example, consider a suboptimal solution wherein two nodes are neighbors in terms of the geometric distance but are distant in their relative sequence positions. Under sequence-based deconstruction, such nodes have a low probability of being in the same destruction space. Consequently, if an optimal solution requires these two nodes to be adjacent, the subsequent reconstruction step is unlikely to reconnect them, thereby causing the suboptimal solution to stall in a local optimum.

To overcome this limitation, we propose an insertion-based local reconstruction method that exploits the inherent flexibility of the insertion paradigm through a distance-based deconstruction strategy, which can enable more effective reconstruction and consequently enhance the model performance. The iterative local reconstruction process alternates between destruction and reconstruction phases. **(1) Destruction Phase:** The solution is destroyed based on the distance-based proximity of nodes. The process begins by randomly selecting a node from the current solution, followed by the identification and removal of its $\alpha$-nearest neighbors, which transition to unvisited nodes. The remaining solution segments automatically form a new reduced loop solution. Here, $\alpha$ represents the destruction size. For example, given a TSP10 solution $(\pi_1, \pi_2, \ldots, \pi_{10})$ where $\pi_5$ is randomly selected with its 3-nearest neighbors $\{\pi_2, \pi_7, \pi_8\}$, the removal of $\pi_5$ and $\{\pi_2, \pi_7, \pi_8\}$ yields the looped solution $(\pi_1, \pi_3, \pi_4, \pi_6, \pi_9, \pi_{10})$. **(2) Reconstruction Phase:** The model sequentially inserts the unvisited nodes into this partial solution. Once all nodes are re-inserted, the newly generated solution is compared against its predecessor, and the superior solution is retained for subsequent iterations. This iterative process continues until a predefined computational budget (e.g., a maximum number of iterations) is exhausted.

## 4 Experiments

In this section, we empirically evaluate the L2C-Insert framework on both synthetic and real-world benchmark TSP and CVRP datasets of various scales. We then contact ablation studies to verify the effect of the proposed framework's critical components.

**Datasets** Following the standard data generation procedure from prior work [19], we generate datasets for both TSP and CVRP problems. **(1) Training Datasets:** For each problem (TSP and CVRP), we create training datasets consisting of one million 100-node instances. We obtain (near-)optimal solutions for these instances using Concorde [41] for TSP and HGS [42] for CVRP. **(2) Test Datasets:** Consistent with established practices [43, 22, 44, 45], we generate five synthetic test datasets with instance sizes ranging from 100 to 100K nodes (100, 1K, 10K, 50K, and 100K). We denote TSP and CVRP instances with $N$ nodes as TSP$N$ and CVRP$N$ (e.g., TSP100, CVRP100). The dataset composition is as follows: For TSP, the TSP100 and TSP1K datasets contain 10,000 and 128 instances, respectively, while larger-scale datasets (10K, 50K, 100K nodes) each have 16 instances. CVRP datasets follow a similar structure, with the exception that CVRP1K contains 100 instances, as specified in [44]. Following Hou et al. [44], we set vehicle capacities to 50 for CVRP100, 200 for CVRP1K, and 300 for larger-scale instances. To evaluate our method on real-world large-scale instances, we also collect 49 symmetric TSP instances and 100 CVRP instances with EUC_2D features and $\leq$1K nodes from TSPLIB [46] and CVRPLIB Set-X [47], respectively.

**Model Setting & Training** The embedding dimension is set to $d = 128$, the hidden dimension of the feed-forward layer is set to 512, the query dimension $d_q$ is set to 16, and the head number in multi-head attention is set to 8. We set the number of attention layers in the decoder to $L = 9$ following [23]. Both the TSP and CVRP models are trained on one million instances of size 100. For

Table 1: Results on synthetic TSP and CVRP instances. Results marked with an asterisk (*) are from the original paper. The best results are bold. OOM: The method triggered an out-of-memory condition. The reported inference time is the total time required to solve the entire test set.

| Method | TSP100 | | | TSP1K | | | TSP10K | | |
| --- | --- | --- | --- | --- | --- | --- | --- | --- | --- |
| | Length | Gap | Time | Length | Gap | Time | Length | Gap | Time |
| LKH3 | 7.76325 | 0.000% | 56m | 23.119 | 0.000% | 8.2h | 71.778 | 0.000% | 8.8h |
| Concorde | 7.76324 | 0.000% | 34m | 23.118 | -0.003% | 7.8h | 72 | 0.310% | 22.4h |
| Att-GCN+MCTS* | 7.764 | 0.037% | 15m | 23.863 | 3.224% | 13m | 74.93 | 4.390% | 1.8h |
| DIFUSCO* | 7.78 | 0.240% | - | 23.56 | 1.900% | 12.1m | 73.62 | 2.580% | 47m |
| T2T* | 7.76 | 0.060% | - | 23.3 | 0.780% | 54.7m | - | - | - |
| NeurOpt | 7.765 | 0.023% | 1.8h | 307.578 | > 100% | 3h | - | - | - |
| SO-mixed* | - | - | - | 23.766 | 2.800% | 55.5m | 74.299 | 3.517% | 2.034h |
| H-TSP | - | - | - | 24.718 | 6.912% | 34s | 77.75 | 8.320% | 25m |
| GLOP | 7.767 | 0.048% | 1.4h | 23.84 | 3.119% | 2.4m | 75.04 | 4.545% | 25m |
| UDC-$\mathbf{x}_{50}$ ($\alpha = 50$) | 7.788 | 0.319% | 1.38h | 23.53 | 1.782% | 6.21m | OOM | OOM | OOM |
| POMO aug$\times$8 | 7.774 | 0.138% | 0.7m | 32.5 | 40.577% | 6.1m | OOM | OOM | OOM |
| INViT-3V aug$\times$16 | 7.92 | 2.019% | 3.02m | 24.343 | 5.299% | 30.43m | 76.494 | 6.570% | 1.19h |
| ELG | 7.781 | 0.229% | 2.2m | 25.738 | 11.328% | 1.3m | OOM | OOM | OOM |
| BQ bs16 | 7.764 | 0.010% | 20m | 23.432 | 1.354% | 21m | OOM | OOM | OOM |
| LEHD RRC1000 | 7.76337 | 0.0016% | 2.3h | 23.288 | 0.731% | 5.2h | 80.9 | 12.709% | 3.6h |
| L2C-Insert Greedy | 7.799 | 0.459% | 0.86m | 24.218 | 4.754% | 0.35m | 77.343 | 7.754% | 1.06m |
| L2C-Insert ($I$=100) | 7.764 | 0.0038% | 47.9m | 23.414 | 1.275% | 4.79m | 75.603 | 5.329% | 2.58m |
| L2C-Insert ($I$=200) | 7.76336 | 0.0014% | 1.62h | 23.339 | 0.952% | 9.52m | 74.733 | 4.117% | 4.14m |
| L2C-Insert ($I$=500) | 7.76328 | 0.0005% | 4.0h | 23.265 | 0.633% | 22.94m | 73.812 | 2.834% | 8.73m |
| L2C-Insert ($I$=1000) | **7.76326** | **0.0002%** | 8.0h | **23.230** | **0.481%** | 46.4m | **73.270** | **2.079%** | 16.64m |

| Method | CVRP100 | | | CVRP1K | | | CVRP10K | | |
| --- | --- | --- | --- | --- | --- | --- | --- | --- | --- |
| | Length | Gap | Time | Length | Gap | Time | Length | Gap | Time |
| HGS | 15.56 | 0.000% | 4.5h | 41.161 | 0.000% | 5h | 226.177 | 0.000% | 8h |
| LKH3 | 15.65 | 0.540% | 12h | 42.156 | 2.416% | 6.1h | 290.575 | 28.472% | 10.3h |
| NeurOpt | 15.66 | 0.627% | 3h | - | - | - | - | - | - |
| GLOP-G (LKH3) | - | - | - | 45.753 | 11.154% | 1.93m | 276.252 | 22.140% | 27.98m |
| UDC-$\mathbf{x}_{50}$ ($\alpha = 50$) | 16.291 | 4.699% | 2.69m | 43.583 | 5.884% | 11.96m | OOM | OOM | OOM |
| POMO aug$\times$8 | 15.75 | 1.230% | 0.86m | 100.990 | >100% | 5.83m | OOM | OOM | OOM |
| INViT-3V aug$\times$16 | 16.538 | 6.282% | 3.93m | 46.607 | 13.230% | 26.97m | 275.691 | 21.892% | 1.28h |
| ELG | 15.84 | 1.760% | 3.4m | 46.847 | 13.812% | 3.16m | 277.685 | 22.773% | 11.92m |
| BQ bs16 | 15.81 | 1.560% | 18m | 43.121 | 4.762% | 15m | 275.494 | 21.805% | 1.36h |
| LEHD RRC1000 | 15.63 | 0.420% | 2.0h | **42.408** | **3.028%** | 4.15h | 248.604 | 9.916% | 8.1h |
| L2C-Insert Greedy | 16.169 | 3.892% | 1.4m | 44.458 | 8.009% | 1.17m | 255.356 | 12.901% | 2.88m |
| L2C-Insert ($I$=100) | 15.700 | 0.877% | 1.0h | 43.868 | 6.574% | 12.08m | 253.462 | 12.063% | 5.75m |
| L2C-Insert ($I$=200) | 15.667 | 0.664% | 2.0h | 43.677 | 6.113% | 24.66m | 252.021 | 11.426% | 9.02m |
| L2C-Insert ($I$=500) | 15.640 | 0.493% | 5.3h | 43.416 | 5.478% | 1.05h | 248.987 | 10.085% | 19.03m |
| L2C-Insert ($I$=1000) | **15.627** | **0.413%** | 10.6h | 43.152 | 4.836% | 2.05h | **245.732** | **8.646%** | 34.72m |

TSP model training, we employ the Adam optimizer [37]. For CVRP model training, we employ the AdamW [48] optimizer, as we find it to be more stable. The initial learning rate is $1\times10^{-4}$, decaying by a factor of 0.97 after each epoch. The TSP and CVRP models undergo 50 and 15 training epochs, respectively, with a batch size of 1024. All experiments, including training, testing, and evaluation, are conducted on a single NVIDIA GeForce RTX 4090 GPU with 24 GB of memory to ensure consistent computational conditions.

**Baselines** We compare L2C-Insert with a comprehensive set of baseline methods including **(1) Classical Solvers:** Concorde [41], LKH3 [49], and HGS [42]; **(2) Constructive NCO Methods:** POMO [21]; BQ [23], LEHD [22], INViT [50], and ELG [30]. (3) **Heatmap-based NCO Methods:** Att-GCN+MCTS [43], DIFUSCO [51], and T2T [52]; (4) **Two-stage NCO Methods:** H-TSP [53], SO-mixed [54], GLOP [40], and UDC [40]; (5) **Improvement-based NCO Methods:** NeurOpt [55]. In line with the established practice in the NCO literature [19, 21], the classical solvers are primarily included to provide a strong, near-optimal baseline for solution quality, rather than to serve as direct competitors in terms of runtime. We refer to Appendix C for implementation details of these baselines.

Table 2: Results on synthetic TSP and CVRP instances with 50K/100K nodes. The best results are bold.

| Method | TSP50K | | | TSP100K | | |
|---|---|---|---|---|---|---|
| | Length | Gap | Time | Length | Gap | Time |
| LKH3 | 159.93 | 0.000% | 160h | 225.99 | 0.000% | 400h |
| GLOP | 168.09 | 5.102% | 24m | 237.61 | 5.142% | 1.04h |
| INViT greedy | 171.42 | 7.184% | 20.8h | 242.26 | 7.199% | 80h |
| LEHD RRC1000 | OOM | OOM | OOM | OOM | OOM | OOM |
| L2C-Insert Greedy | 171.778 | 7.408% | 5.28m | 242.757 | 7.419% | 10.53m |
| L2C-Insert ($I$=200) | 169.802 | 6.173% | 8.35m | 241.194 | 6.728% | 13.60m |
| L2C-Insert ($I$=1000) | **166.063** | **3.835%** | 20.83m | **237.109** | **4.920%** | 26.07m |

| Method | CVRP50K | | | CVRP100K | | |
|---|---|---|---|---|---|---|
| | Length | Gap | Time | Length | Gap | Time |
| HGS | 1081.0 | 0.000% | 64h | 2087.5 | 0.000% | 100.8h |
| INViT-3V greedy | 1331.1 | 23.136% | 46.4h | 2683.4 | 28.546% | 93h |
| BQ bs16 | OOM | OOM | OOM | OOM | OOM | OOM |
| LEHD RRC1000 | OOM | OOM | OOM | OOM | OOM | OOM |
| L2C-Insert Greedy | 1188.084 | 9.906% | 14.49m | 2271.292 | 8.804% | 29.95m |
| L2C-Insert ($I$=200) | 1184.744 | 9.597% | 20.90m | 2268.146 | 8.654% | 37.67m |
| L2C-Insert ($I$=1000) | **1174.208** | **8.622%** | 47.63m | **2258.005** | **8.168%** | 1.16h |

Table 3: Comparison result for appending- and insertion-based constructive heuristics on TSPLIB and CVRPLIB instances.

| Method | TSPLIB | | | CVRPLIB | | |
|---|---|---|---|---|---|---|
| | N<=200 | 200<N<=1K | All | N<=200 | 200<N<=1K | All |
| NeurOpt | 9.456% | 26.345% | 16.350% | 24.878% | 25.301% | 25.190% |
| UDC-$\mathbf{x}_{250}$ ($\alpha = 50$) | 0.533% | 6.578% | 3.072% | 9.635% | 8.864% | 9.033% |
| GLOP | 0.561% | 2.012% | 1.153% | - | - | - |
| INViT-3V aug×16 | 2.238% | 5.654% | 3.673% | 8.231% | 11.504% | 10.784% |
| ELG | 1.180% | 5.910% | 3.111% | 4.510% | 7.752% | 7.039% |
| POMO aug×8 | 2.022% | 21.447% | 9.951% | 6.869% | 46.984% | 38.159% |
| BQ bs16 | 1.289% | 1.946% | 1.557% | 7.661% | 8.195% | 8.078% |
| LEHD RRC1000 | 0.215% | 0.620% | 0.380% | 2.022% | 3.932% | 3.512% |
| L2C-Insert ($I$=1000) | **0.197%** | **0.468%** | **0.308%** | **1.604%** | **3.651%** | **3.201%** |

**Inference & Metrics** For inference, the L2C-Insert model first generates an initial solution through greedy rollout, then progressively refines solution quality via insertion-based local reconstruction. The number of iterations is denoted as "$I$=". Following [23], we accelerate inference by processing only the $k$-nearest unvisited nodes and positions of the current node, with $k$ set to 100 for TSP and 200 for CVRP. To further accelerate this process, we limit the local reconstruction destruction size $\alpha$ to 300 across all instances. Following [40], to enhance the consistency of inputs for our model on large-scale instances with more than 1K nodes, we employ min-max feature scaling, which adjusts the x-axis and y-axis coordinate values to the interval [0, 1]. A hyperparameter study and discussion on these settings are provided in Appendix D. For performance evaluation, we measure: 1) average tour length (Length), 2) performance gap relative to baseline solvers (Gap), and 3) total inference time (Time). Note that reported inference times for classical CPU-based solvers are not directly comparable to GPU-executed learning-based methods.

## 4.1 Comparative Results

Tables 1 and 2 present the comparative results on uniformly distributed TSP and CVRP instances scaling from 100 to 100K nodes. These results show that L2C-Insert consistently achieves outstanding performance across these scales with a reasonable runtime. On TSP100, L2C-Insert ($I$=200) reduces the gap of the advanced NCO baseline LEHD+RRC1000 by 12.5% with a speedup of $1.4\times$. On TSP1K, L2C-Insert ($I$=1000) achieves a significant reduction of 34.2% while with a speedup of $6.7\times$ compared to LEHD+RRC1000. On TSP10K, L2C-Insert ($I$=1000) outperforms the strong heatmap-based solver DIFUSCO and two-stage solver GLOP with 19.4% and 54.3% gap reduction, while with $2.8\times$ and $1.5\times$ speedup, respectively. On TSP50K and TSP100K, L2C-Insert ($I$=200) outperforms the two-stage NCO method GLOP with a gap reduction of 24.8% and 4.3%, respectively.

On CVRP100, L2C-Insert ($I$=1000) outperforms LEHD RRC1000 by 1.7%, and on CVRP1K it achieves a competitive gap while with a speedup of 2×. On CVRP10K, L2C-Insert ($I$=1000) outperforms LEHD RRC1000 with a speed-up of 14×. On CVRP50K and CVRP100K, L2C-Insert using greedy rollout achieves significant gap reductions of 62.7% and 71.4%, respectively, compared to INViT. Overall, our method shows good scalability and can achieve outstanding performance across small-scale to very large-scale problem instances with up to 100K nodes. The experimental results on real-world TSPLIB and CVRPLIB instances are provided in Table 3. These results show that L2C-Insert consistently outperforms representative neural solvers on all instance groups, demonstrating robust generalization ability.

## 4.2 Ablation Study

**Learning to Append vs. Learning to Insert**    We conduct a comparative evaluation between two constructive NCO approaches: learning to append (L2C-Append) and learning to insert (L2C-Insert). While L2C-Append shares a similar model architecture with L2C-Insert, it utilizes an appending construction framework akin to LEHD [22]. Both models are trained on TSP100 under identical budgets and subsequently evaluated on TSP100, TSP1K, and TSP10K datasets. For inference, we apply local reconstruction with 200 and 1000 iterations using the aforementioned settings. As detailed in Table 4, L2C-Insert, with only 200 reconstruction iterations, outperforms L2C-Append even when the latter uses 1000 iterations of local reconstruction while also consuming less runtime. Remarkably, L2C-Insert ($I$=1000) outperforms L2C-Append ($I$=1000) by achieving substantial gap reductions of at least 50.6% on these datasets. These findings highlight that the L2C-Insert framework's greater flexibility in solution construction directly contributes to its superior performance.

Table 4: Appending-based Construction vs. Insertion-based Construction. The best gap is indicated in bold, and the second-best is underlined.

| Method | | TSP100 | | TSP1000 | | TSP10000 | |
|---|---|---|---|---|---|---|---|
| | | Gap | Time | Gap | Time | Gap | Time |
| L2C-Append | ($I$=200) | 0.00489% | 28.03m | 1.159% | 3.49m | 22.986% | 4.11m |
| | ($I$=1000) | 0.00144% | 2.28h | 0.974% | 16.88m | 12.881% | 16.30m |
| L2C-Insert | ($I$=200) | 0.00142% | 1.62h | 0.952% | 9.52m | 4.117% | 4.14m |
| | ($I$=1000) | **0.00017%** | 8h | **0.481%** | 46.4m | **2.079%** | 16.64m |

**Effect of Destruction Strategy**    We ablate the effect of distance-based strategies on local reconstruction. Specifically, on TSP1K and TSP10K, we test the L2C-Insert model using local reconstruction ($I$=1000) with the distance-based and sequence-based destruction separately. As demonstrated in Table 5, the distance-based destruction strategy achieves substantially better performance, indicating its greater effectiveness in helping local reconstruction escape local optima.

Table 5: Effect of deconstruction strategy.

| Method | TSP1000 Gap | TSP10000 Gap |
|---|---|---|
| Sequence-based | 1.182% | 6.980% |
| Distance-based | **0.481%** | **2.079%** |

**Additional Analysis & Discussion**    We further investigate the impact of: 1) different distinct distance metrics (Euclidean vs. Polar Angle); 2) the use of different node selection strategies (Random vs. Nearest); 3) the incorporation of Positional Encodings [36] into embeddings of positions within the partial solution; 4) the exclusion of unvisited nodes as decoder input; and 5) the use of different backbone models for L2C-Insert framework. Detailed results and the corresponding discussion are provided in Appendix E.

## 5   Conclusion, Limitation, and Future Work

**Conclusion**    In this work, we have proposed L2C-Insert, a novel learning to insert framework to explore the potential of the insertion paradigm for constructive NCO. By strategically inserting unvisited nodes at any valid position within the partial solution, L2C-Insert can flexibly construct high-quality solutions for routing problems. Extensive comparisons with advanced neural solvers demonstrate L2C-Insert's outstanding performance on TSP and CVRP.

**Limitation and Future Work** First, although L2C-Insert has demonstrated outstanding performance, its performance could be further boosted through the development of more effective model architectures. These improved architectures would better capture the intricate relationships between current nodes, unvisited nodes, and positions within partial solutions. Second, designing a hybrid NCO solver that combines both appending and inserting moves is a highly promising research direction. Such a model has the potential to be more flexible and powerful than a model restricted to a single type of move. In addition, a limitation may be the greedy, non-learned nearest-neighbor heuristic used for node selection. A promising research direction is to develop a learned policy that works synergistically with the insertion model, potentially through joint or two-stage training. Finally, extending the L2C-Insert framework to address a broader range of combinatorial optimization problems is another promising research direction for future work.

## Acknowledgments and Disclosure of Funding

This work was supported in part by the National Natural Science Foundation of China (Grant 62476118, Grant 12426305), the Natural Science Foundation of Guangdong Province (Grant 2024A1515011759), the Guangdong Science and Technology Program (Grant 2024B1212010002), the Center for Computational Science and Engineering at Southern University of Science and Technology, the National Key Research and Development Program of China (Grant 2022YFA1004201), and the Research Grants Council of the Hong Kong Special Administrative Region, China (GRF Project No. CityU 11212524).

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

# A  Related Work

## A.1  Constructive NCO Methods

Constructive NCO methods typically arrange the input nodes into high-quality solutions autoregressively. This paradigm is inspired by the sequence-to-sequence framework [27] in natural language processing (NLP), where a model encodes an input sequence into an embedding and then autoregressively generates an output sequence based on this representation. Vinyals et al. [6] pioneer this adaptation through their Pointer Networks (Ptr-Nets), which establishes the first sequence-to-sequence framework specifically for neural combinatorial optimization. In this paradigm, the neural model learns to predict the probability of appending the unvisited nodes to the partial solution until all nodes are visited. We call this paradigm Learning to Append (L2C-Append). Subsequent research has further advanced L2C-Append NCO methods, with improvements focusing on model training techniques [7], extension to solve other COPs Nazari et al. [56], and architectural enhancements [19]. Among them, the Attention Model [19] introduces the Transformer-based model structure [36], which achieves outstanding performances over various VRPs with up to 100 nodes. Various AM-based improved variants are proposed to narrow the performance gap to classical heuristic solvers [57, 21, 58–64, 22, 23, 65, 39, 66, 40, 67–69, 26, 70, 71]. For example, PolyNet [33] enhances exploration by learning a diverse set of complementary solution strategies in a single model. By conditioning the solution generation process on a simple bit vector, PolyNet can generate diverse solutions without relying on handcrafted rules or forced initial moves. Nonetheless, L2C-Append NCO methods are fundamentally limited by their restriction to appending new nodes, which can lead to suboptimal solutions due to early greedy choices. In contrast, the insertion-based paradigm offers greater flexibility for VRPs by sequentially inserting unvisited nodes into any valid position [28, 29], potentially yielding higher-quality solutions. Despite these benefits, the insertion-based constructive approach has remained unexplored in NCO for VRPs in the last decade. In this work, we introduce L2C-Insert, a novel learning to insert framework, to investigate the potential of this paradigm for constructive NCO.

## A.2  Non-constructive NCO Methods

This series of methods can be broadly categorized into three main groups: **(1) Heatmap-based methods**: These methods learn graph neural networks to predict a probability distribution, or heatmap, indicating the likelihood of each edge being part of the optimal solution for a given problem instance. Once the heatmap is obtained, it is used to guide subsequent decoding techniques such as greedy search, beam search [72], Monte Carlo Tree Search (MCTS) [43, 73, 74], 2-opt [51, 52], or gradient search [75] to generate the solution. While this approach can successfully solve TSP instances of various sizes, its application to other routing problems like CVRP is generally more challenging. **(2) Improvement-based methods**: These approaches iteratively refine solution quality. They employ neural models to predict regions within current solutions where heuristic operators, such as k-opt [55] and swap [76], can be applied to improve the solution quality [77–81, 38]. (3) **Two-stage Method**: Two-stage methods: These methods address the problem through an iterative divide-and-conquer process. In each iteration, the problem is first decomposed into multiple subproblems using rule-based techniques or neural models. Subsequently, neural solvers (e.g., AM [19]) or classical solvers (e.g., LKH3 [49] or HGS [42]) are used to solve these subproblems. The resulting partial solutions are then merged to form a complete solution [82, 80, 44, 32, 40]. Beyond these primary categories, other non-constructive NCO approaches include learning to augment classical solvers with neural components [83] or solving VRPs at a route level [84]. A different approach is Neural Deconstruction Search (NDS) [85]. NDS uses a learned neural policy to iteratively deconstruct a solution by selecting customers to remove. These customers are then reinserted one by one using a rule-based greedy heuristic algorithm. In this paper, we develop a constructive neural model that learns the insertion-based heuristic, building solutions autoregressively.

# B  Implementation Details of L2C-Insert on CVRP

**Problem Formulation**   A CVRP instance $S$ can be represented as a graph with one depot node and $n$ customer nodes, and a vehicle with a fixed capacity $C$. Each node $i \in \{0, 1, \ldots, n\}$ is associated with a feature vector $\boldsymbol{s}_i \in \mathbb{R}^3$, representing its two-dimensional coordinates and demand. Node 0 is the depot, and its demand is zero.

For any given CVRP instance, the solution is a set of routes for the vehicle to serve all customers, such that: (1) Each customer is visited exactly once. (2) Each route must start from and end with the depot. (3) The total demand on each route does not exceed the vehicle's capacity. The objective of solving CVRP is to determine the optimal solution such that the total cost of the routes, typically the total distance traveled, is minimized.

**Solution Construction Process**   Initially, all customer nodes are unvisited. In the first step, the depot node and its nearest unvisited node initialize the incomplete solution. We call the node selected in the last step as **last node** and the node selected in the current step as **current node**. In each step except the first step, only the unvisited customer nodes can be selected as the current node. In each following step, the model predicts both the probability of 1) inserting the current node for each available position in the incomplete solution and 2) inserting it into the middle of the two copies of the depot nodes to initiate a new route.

**Remaining Capacity**   During solution construction, the remaining capacity associated with each route is calculated by using the vehicle's capacity minus the total demand of this route so far.

## B.1  Encoder

The encoder transforms each node feature $\boldsymbol{s}_i \in \mathbb{R}^3$ to node embeddings $\mathbf{h}_i \in \mathbb{R}^d$ by a linear projection:

$$\mathbf{h}_i^0 = \boldsymbol{s}_i W^{(0)} + \mathbf{b}^{(0)} \quad \forall i = 0, 1, \ldots, n, \tag{4}$$

where $W^{(0)} \in \mathbb{R}^{3 \times d}, \mathbf{b}^{(0)} \in \mathbb{R}^d$ are learnable parameters. These initial embeddings are subsequently updated through the attention layer [21, 22] to produce the refined embeddings $\mathbf{h}_i \in \mathbb{R}^d$, which is the output of the encoder.

## B.2  Decoder

At construction step $t$, the decoder receives the embeddings of all nodes, denoted as $\mathbf{h}_i \in \mathbb{R}^d, i = 1, \ldots, n$, the current partial solution $(0, \pi_1, \pi_2, \ldots, 0, \cdots, \pi_{t-1}, 0)$ (Node 0 denotes the depot), and the set of unvisited nodes $\{u_j, u_j \notin \{\pi_{1:t-1}\}\}$. To compute the probability of insertion at the current step, embeddings for three distinct inputs are required: 1) the current node (the node to be inserted), 2) unvisited nodes, and 3) the positions in the partial solution.

**Embeddings of Positions in the Current Partial Solution**   The unvisited node with the minimum Euclidean distance to the last node is selected as the current node to be inserted, of which the embedding is denoted as $\mathbf{h}_{u_c}$. Secondly, the embeddings of the remaining unvisited nodes can be denoted as $H_u = \{\mathbf{h}_{u_j}\}, u_j \notin \{\pi_{1:t-1}\}, u_j \neq u_c$. Two learnable linear projections $W_0 \in \mathbb{R}^{d \times d}$ and $W_1 \in \mathbb{R}^{d \times d}$ are applied to the current node's embedding and each unvisited node's embedding, respectively

**Embeddings of the Current Node and Unvisited Nodes**   To generate the embedding of a potential insertion position within the partial solution, the embeddings of adjacent visited nodes are horizontally concatenated. For example, for a partial solution:

$$0 \rightarrow 1 \rightarrow 2 \rightarrow 0 \rightarrow 4 \rightarrow 5 \rightarrow 6 \rightarrow 0 \rightarrow 7 \rightarrow 8 \rightarrow 0$$

where the positions are indicated by these arrows. The arrow from the tail to the head ($0 \leftarrow 0$) indicates the position that allows the current node to be inserted to start a new route. To simplify the notation, we assume the current incomplete solution is of length $l$, and the embeddings of the nodes

in it are $\{\mathbf{h}_i\}, i \in \{1 : l\}$. Then the set of embeddings of the positions in the partial solution can be denoted as $\{\mathbf{e}_i\}, i \in \{1 : l\}$, where

$$\mathbf{e}_i = \begin{cases} [\mathbf{h}_i, \mathbf{h}_{i+1}] & i \in \{1 : l-1\}, \\ [\mathbf{h}_l, \mathbf{h}_1], & i = l, \end{cases} \tag{5}$$

where $[\cdot, \cdot]$ is the horizontal concatenation operator. We note that both $\mathbf{h}_1$ and $\mathbf{h}_l$ are the depot embeddings. To **incorporate the remaining capacity information**, the decoder updates each position embedding by horizontally concatenating it with the remaining capacity of the route where the position is located, increasing the embedding dimension by 1. Subsequently, a linear projection $W_2 \in \mathbb{R}^{(2d+1) \times d}$ is applied to the updated position embedding to project it to the embedding with the same dimension as the node embedding. The set of output position embeddings is denoted as $H_e = \{\mathbf{e}_{\pi_i}\}, i \in \{1 : t-1\}$.

Then, these embeddings together are refined by $L$ stacked attention layers. The first layer receives input $X^0 = \{\mathbf{h}_c, H_e, H_u\}$, with each subsequent $j$-th attention layer processing the output from the $(j-1)$-th layer, culminating in the final output $X^{(L)}$. This process can be denoted as:

$$\begin{aligned} H^{(1)} &= \text{AttnLayer}(H^{(0)}), \\ H^{(2)} &= \text{AttnLayer}(H^{(1)}), \\ &\cdots \\ H^{(L)} &= \text{AttnLayer}(H^{(L-1)}), \end{aligned} \tag{6}$$

Where the formulation of "AttnLayer" is detailed in Appendix N. Then, a linear projection and softmax function are applied to $X^{(L)}$ to compute the insertion probabilities of the current node into all valid positions. The masking operation assigns $-\infty$ to invalid positions (those corresponding to the current node and non-visited nodes) to ensure proper probability normalization over valid insertion locations. The inserting probability $p_i$ is calculated as:

$$\begin{aligned} x_i &= X_i^{(L)} W_f + b_f, \\ a_i &= \begin{cases} x_i & \forall i \in \{1 : l\} \text{ and the remaining capacity of the route where the position is located} \\ & \text{is not less than the current node's demand,} \\ -\infty & \text{otherwise,} \end{cases} \\ \mathbf{p} &= \text{Softmax}(\mathbf{a}), \end{aligned} \tag{7}$$

where $W_f \in \mathbb{R}^{d \times 1}$ and $b_f \in \mathbb{R}^1$ are learnable parameters. where $W_f \in \mathbb{R}^{d \times 1}$ and $b_f \in \mathbb{R}^1$ are learnable parameters. Each $p_i \in \mathbf{p}$ denotes the probability of inserting the current node between consecutive nodes $(\pi_i, \pi_{i+1})$ in the partial solution. The decoder executes this sampling-and-insertion procedure $n$ times to construct the complete solution $\{0, \pi_1, \cdots, 0, \cdots, \pi_n\}$.

### B.3 Implementation of Insertion-based Local Reconstruction on CVRP

In the main paper, we introduce the process of insertion-based local reconstruction on TSP. This process remains nearly the same as in CVRP, which is described as follows:

- **Solution Initialization** The initial solution is also generated by the model through a simple greedy rollout.

- **Distance-based Destruction** In the CVRP case, we also destroy the solution by the node's neighbor relationship in terms of distances. Specifically, we first randomly sample a customer node. Then, we select its $\alpha$-nearest neighbor customer nodes and remove them as unvisited nodes, the left solution automatically creates an incomplete solution with multiple looped routes. Here, we refer to $\alpha$ as the destruction size. For example, if a CVRP10 solution is $(0, \pi_1, \pi_2, 0, \pi_3, \pi_4, \pi_5, 0, \pi_6, \pi_7, 0, \pi_8, \pi_9, \pi_{10}, 0)$, the node $\pi_5$ is randomly sampled, and its 3-nearest neighbors are $\{\pi_1, \pi_3, \pi_6\}$, then they are removed from the solution to serve as unvisited nodes. The left incomplete solution is $(0, \pi_2, 0, \pi_4, 0, \pi_7, 0, \pi_8, \pi_9, \pi_{10}, 0)$.

- **Insertion-based Reconstruction** The model inserts the unvisited nodes to the incomplete solution node by node. Initially, the last node is randomly selected from the incomplete

solution. The subsequent procedures are the same as those for a greedy rollout. When all nodes are visited, a new complete solution is generated. We compare the previous solution with the new one, accepting the higher-quality solution for the next destruction-reconstruction cycle. This iterative process continues until the computational budget (e.g., maximum iterations or time) is reached.

**Padding for Solution Misalignment**   Since the solution to each CVRP instance may involve a different number of depot visits, the resulting solution lengths can vary when solving multiple CVRP instances in batches. To address this issue, we apply a simple padding technique by adding extra zeros at the end of solutions with fewer depot visits. These extra zero pairs are then masked in the model.

## C   Implementation Details of Baselines

We compare L2C-Insert with a comprehensive set of baseline methods including **(1) Classical Solvers:** Concorde [41], LKH3 [49], and HGS [42]; **(2) Constructive NCO Methods:** POMO [21]; BQ [23], LEHD [22], INViT [50], and ELG [30].  (3) **Heatmap-based NCO Methods:** Att-GCN+MCTS [43], DIFUSCO [51], and T2T [52]; (4) **Two-stage NCO Methods:** H-TSP [53], SO-mixed [54], GLOP [40], and UDC [40]; (5) **Improvement-based NCO Methods:** NeurOpt [55].

For the heatmap-based methods, we directly use their reported results. For SO-mixed, as their source codes are not released available, we choose to directly report their original results. However, we use the same test datasets for fair comparison (all these methods used the same TSP1K and TSP10K test datasets from Fu et al. [43]). For other methods, if not otherwise stated, we directly run their source codes with the default settings.

- **Conconde** [41] We run the Python wrapper for the concoder, namely PyConcorde. We use the default settings across TSP instances of various sizes.
- **LKH3** [49] We run the code provided by AM [19]. For TSP100K, we use POPMUSIC [86] to reduce the complexity for pre-processing (by setting CANDIDATE SET TYPE = POP-MUSIC and INITIAL PERIOD=1000), following the suggestion from Fu et al. [87]. Other settings are default.
- **HGS** [42] We run the Python wrapper for the HGS-CVRP solver, namely PyHygese.
- **LEHD** [22] We limited the maximum destruction size of RRC to 1000 after observing that performance remained nearly unchanged when compared to cases without this restriction, while inference time increased significantly.

# D   Hyperparameter Studies

We conduct hyperparameter studies for the values of neighborhood size $k$ and destruction size $\alpha$ by running L2C-Insert using local reconstruction with $I$=50 iterations.

## D.1   Defining the Neighbor Unvisited Nodes and Positions

**Neighbor Unvisited Node**   Given a current node and a unvisited node-set $B$, we categorize the unvisited nodes in $B$ that are closer to the current node as neighbor nodes based on the distance.

**Neighbor Position**   The position-node distance between the position and current node is measured by the smallest distance between the current node and the adjacent visited nodes of this position. Given a current node and a position set $E$, we categorize the positions in $E$ that are closer to the current node as neighbor positions based on the position-node distance.

## D.2   On TSP

As detailed in Table 6, model performance initially improves with an increase in destruction size from 100 to 300, beyond which it declines as the destruction size extends to 500. Consequently, we select an intermediate destruction size of 300 to achieve optimal performance. Furthermore, our analysis reveals that increasing the neighborhood size from 100 to 500 leads to progressively longer solution lengths. We hypothesize that the intricate interplay of numerous positions and nodes complicates the processing, thereby diminishing the model's effectiveness. Therefore, we set the neighborhood size to 100 to enable robust problem-solving while maintaining a low runtime.

Table 6: Effect of the setting of neighborhood $k$ and destruction size $\alpha$ in inference stage on TSP10K.

| | | Destruction Size $\alpha$ | | |
| | | 100 | 300 | 500 |
|---|---|---|---|---|
| | | Length (Time) | Length (Time) | Length (Time) |
| Neighborhood Size $k$ | 100 | 76.669 (1.32m) | **76.573 (2.04m)** | 77.078 (3.94m) |
| | 200 | 78.407 (1.68m) | 76.610 (3.11m) | 78.320 (5.77m) |
| | 300 | 80.349 (2.26m) | 78.983 (4.17m) | 79.042 (7.29m) |
| | 500 | 81.448 (4.36m) | 80.451 (7.48m) | 79.150 (12.18m) |

## D.3   On CVRP

As indicated in Table 7, model performance generally improves with increasing destruction size from 100 to 300, but declines when the destruction size further extends from 300 to 500. Consequently, we select an intermediate destruction size of 300 to achieve favorable results. Furthermore, with the destruction size set at 300, the model exhibits significantly better performance when the neighborhood size is 200, leading us to choose this value for our experiments.

Table 7: Effect of the setting of neighborhood $k$ and destruction size $\alpha$ in inference stage on CVRP1K.

| | | Destruction Size $\alpha$ | | |
| | | 100 | 300 | 500 |
|---|---|---|---|---|
| | | Length (Time) | Length (Time) | Length (Time) |
| Neighborhood Size $k$ | 100 | 43.011 (0.51m) | 43.196 (0.71m) | 43.530 (1.31m) |
| | 200 | 43.127 (0.95m) | **42.783 (1.53m)** | 43.069 (2.78m) |
| | 300 | 43.030 (1.29m) | 42.989 (2.15m) | 43.166 (3.81m) |
| | 500 | 43.178 (2.53m) | 42.887 (4.37m) | 43.204 (7.47m) |

# E Additional Ablation Studies and Analysis

Each step in the insertion-based construction process is divided into two substeps: (1) selecting a node from the unvisited nodes and (2) selecting an edge in the incomplete solution for insertion. In this paper, the second substep is conducted by the model, and the first substep is operated through predetermined rules. Specifically, for the first substep, we define the insertion node selected in the previous step as the last node, and the insertion node selected in the current step as the current node. The current node is the unvisited node closest to the last node. In this section, we study the impact of the distance criterion and the current node selection strategy on model performance. Beyond those, we also study the impact of adding the Positional Encoding [36] to the embeddings of the positions in the partial solution, the effect of unvisited nodes as decoder input, and the performance of L2C-Insert using different backbone models.

## E.1 Effect of Distance Criterion for CVRP

We test two classic distance criteria, Euclidean Distance and Polar Angle. For two customer nodes $\mathbf{p} = (p_1, p_2)$ and $\mathbf{q} = (q_1, q_2)$, Euclidean Distance is defined as: $d_E(\mathbf{p}, \mathbf{q}) = \sqrt{(p_1 - q_1)^2 + (p_2 - q_2)^2}$. For Polar Angle, we must introduce the depot node $\mathbf{d}_c = (d_1, d_2)$ to serve as the Polar center. Then it is defined as: $\theta_P(\mathbf{p}, \mathbf{q}) = |\theta_\mathbf{p} - \theta_\mathbf{q}|$, where $\theta_\mathbf{p} = \arctan\left(\frac{p_2 - d_2}{p_1 - d_1}\right)$ and $\theta_\mathbf{q} = \arctan\left(\frac{q_2 - d_2}{q_1 - d_1}\right)$ are the angles formed by the lines connecting the depot $\mathbf{d}_c$ to the customer nodes $\mathbf{p}$ and $\mathbf{q}$ with respect to the horizontal axis.

We train and test the same model with these two distance criteria, maintaining the node selection strategy that the current node is the closest unvisited node to the last one. Both models are trained with the same budget mentioned in the main paper. We then test these models on CVRP100 and CVRP1000 using a greedy rollout. The results, as shown in Table 8, indicate that the model trained and tested with Polar Angle performs much better. This improvement can be attributed to the nature of CVRP, where multiple routes are usually ordered around a central depot. Using polar angles is tightly coupled to the solution structure of CVRP. As a result, the model is easier to learn and ultimately achieves better performance. Therefore, we use the Polar Angle to measure the neighbor relationship on CVRP.

Table 8: Effect of distance criterion for CVRP.

|  | CVRP100 Gap | CVRP1000 Gap |
|---|---|---|
| Euclidean Distance | 5.151% | 10.155% |
| Polar Angle | **3.892%** | **8.009%** |

## E.2 Effect of Node Select Strategy

We use two different current node selection strategies described in Nilsson [88] for comparison:

- Random: select an unvisited node randomly.
- Nearest Neighbor: select the unvisited node that is nearest to the last node.

Specifically, we train the same models with two different current node selection strategies. The CVRP models use the same training budget mentioned in the main paper, while TSP models are trained with a budget of 10 epochs due to the runtime. We test these models on TSP100 and CVRP100 using greedy rollout. The results in Table 9 show that the nearest neighbor node selection strategy has the best overall result. Therefore, in this paper, we choose to use it for the selection of the current node.

Table 9: Effect of selection strategy.

|  | TSP100 Gap | CVRP100 Gap |
|---|---|---|
| Random | 1.1557% | 11.370% |
| Nearest Neighbor | **0.9535%** | **3.892%** |

### E.3 Effect of Positional Encoding

We compare the performance of models with and without incorporating the Positional Encoding (PE) from [36] to the embeddings of positions in the partial solution. The models are trained on TSP100 with a budget of 10 epochs, and tested on TSP100 using greedy rollout. The results in Table 10 show that incorporating PE will damage both in-domain and out-of-domain performance. Therefore, this paper does not include the positional encoding in our model.

Table 10: Effect of selection strategy.

|        | TSP100
Gap | CVRP100
Gap |
|--------|---------------|----------------|
| W/ PE  | 1.0475%       | 5.802%         |
| W/o PE | **0.9535%**   | **3.892%**     |

### E.4 Effect of Decoder Input of Unvisited Nodes

We compare the performance of models trained on TSP100 with and without inputting unvisited nodes to the decoder. The results in table 11 show that while inputting unvisited nodes to the model results in additional computational cost, the performance improvement is significant, proving its indispensability.

Table 11: Training with unvisited node vs. without.

| Method | TSP100 | | TSP1K | |
|--------|--------|------|--------|-------|
|        | Gap    | Time | Gap    | Time  |
| W/o    | 5.161% | 0.44m | 23.979% | 0.15m |
| W/     | **0.460%** | 0.86m | **4.757%** | 0.36m |

**Why the L2C-Insert Model Necessitates the Unvisited Nodes as the Decoder Input**   To illustrate why this input is crucial, consider the example depicted in Figure 4. Figure 4 (a) displays the optimal solution for a TSP instance, while Figure 4 (b) shows an intermediate state during the solution construction process. Without considering unvisited nodes, the insertion action shown in Figure 4 (c), despite yielding a better partial solution, constitutes a suboptimal choice when evaluated against the global optimum considering all nodes. In contrast, when unvisited nodes are factored in, the insertion action in Figure 4 (d), even though it results in a partial solution with path intersections, is identifiable as a more globally optimal insertion by referencing the overall optimal solution. Therefore, insertions made without information about unvisited nodes tend to be shortsighted. Incorporating unvisited nodes, however, provides the model with a broader context, enabling more informed decisions for selecting insertion positions. Consequently, the L2C-Insert model needs to leverage information from unvisited nodes to mitigate such shortsightedness, thereby guiding the construction process towards higher-quality solutions.

### E.5 Performance of L2C-Insert using Different Backbone Models

We conduct experiments to investigate the performance of the proposed L2C-Insert framework under different backbone NCO models. Specifically, three recently proposed models are employed: POMO [21], LEHD [22], and INVIT [50], which feature 1, 3, and 6 decoder layers, respectively, each within a 7-layer overall model architecture. These models represent light, balanced, and heavy decoder architectures, respectively. Building upon these models, L2C-Insert variants are developed using the proposed L2C-Insert framework. These models are evaluated using the TSPLIB and CVRPLIB instances detailed in Section 4. Both the L2C-Append and L2C-Insert methods utilize a one-shot greedy search for inference.

The experimental results are presented in Table 12. These results indicate that for the LEHD model, the L2C-Insert method outperforms the L2C-Append method on 4 out of 6 instance groups. Conversely, the L2C-Insert method, when based on the POMO model, generally underperforms the L2C-Append method, proving inferior in 3 out of 6 instance groups, respectively. Similarly, the L2C-Insert method, built upon the INViT model, also generally underperforms L2C-Append,

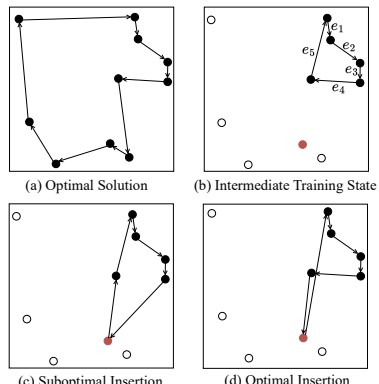

Figure 4: figure

Example of a suboptimal insertion (not considering unvisited nodes) versus an optimal insertion (considering unvisited nodes). The red node is the current node to insert.

showing inferiority in 6 out of 6 instance groups. These findings suggest that the L2C-Insert method may be more effective when utilized with a heavy decoder model. This advantage could stem from the complex position relationships within an incomplete solution and the current node being inserted during solution construction. Capturing these intricate relationships effectively necessitates a heavy decoder with sufficient model capacity to inform appropriate decisions regarding the correct insertion position.

Table 12: Performance of L2C-Insert and L2C-Append using different backbone models. These methods use the greedy rollout (Greedy) for inference.

| Method | | TSPLIB | | | CVRPLIB | | | Perform Better # |
|---|---|---|---|---|---|---|---|---|
| (Using Greedy) | | N<=200 | 200<N<=1K | All | N<=200 | 200<N<=1K | All | |
| POMO | Append | **5.325%** | **35.337%** | **17.575%** | 15.596% | 41.759% | 36.003% | 3/6 |
| | Insert | 19.111% | 38.344% | 26.961% | **14.650%** | **19.305%** | **18.281%** | 3/6 |
| INViT | Append | **2.788%** | **6.315%** | **4.228%** | **8.046%** | **11.313%** | **10.594%** | 6/6 |
| | Insert | 3.961% | 8.107% | 5.653% | 14.927% | 22.551% | 20.874% | 0/6 |
| LEHD | Append | 1.922% | **3.646%** | **2.626%** | 11.346% | 12.851% | 12.520% | 1/6 |
| | Insert | **1.526%** | 4.382% | 2.691% | **10.903%** | **9.548%** | **9.846%** | 4/6 |

# F Discussion on the Advantage of the Insertion-based Construction Paradigm in Search

The iterative local reconstruction is a significant factor in achieving the final reported performance. We believe the key innovation of our work, the insertion-based policy, is what unlocks the true potential of this search procedure. The policy's inherent flexibility is a key enabler for the search's effectiveness.

**1. The advantage of insertion-based search**

As we discuss in Section 3.3, conventional appending-based models face a fundamental limitation when used with local search. When nodes are removed during the destruction phase, they can only be re-appended to the end of the sequence during reconstruction. This rigidity makes it difficult for the model to escape local optima. For instance, reconnecting two nodes that are geometrically close but far apart in the sequence—a common requirement for improving a tour—is highly improbable.

Our L2C-Insert framework directly overcomes this bottleneck. By allowing a node to be re-inserted at any valid position, our model facilitates a much more powerful and effective exploration of the solution space during the reconstruction phase. This flexibility is the core reason why the destroy-and-reconstruct loop is effective in our framework.

### 2. Direct Ablation Study: Isolating the Policy's Contribution

To provide a direct and fair comparison that disentangles the policy's contribution, we conduct the ablation study presented in Table 4 in the main paper. In this experiment, we compared our L2C-Insert model against an L2C-Append model (which uses a similar architecture but an appending policy) within the exact same local reconstruction framework.

The results show that our L2C-Insert with only 200 iterations significantly outperforms the L2C-Append model that uses 1000 iterations. This finding directly demonstrates that the superior performance stems from the intrinsic quality and flexibility of the insertion policy itself, and not just from the application of a search method.

## G    Clarification on Fair Comparison and Calculating Costs

### 1. Clarifying Iteration Settings of Baselines

To clarify, we summarize the iteration settings of iterative methods in the table below. From this table, we can observe that he number of iterations varies significantly. For instance, NeurOpt utilizes 10,000 lightweight k-opt iterations, while two-stage methods like UDC and SO-mixed perform 50-100 divide-and-conquer iterations.

Table 13: Iteration settings for baselilne methods. "Reviser-$n$" refers to a solver designed to repair a local solution of length $n$.

| Method | Problem | Number of iteration |
|---|---|---|
| NeurOpt | TSP & CVRP | 10000 (k-opt) |
| SO-mixed | TSP1K
TSP10K | 50 (Reviser-100/50)
100 (Reviser-100/50) |
| UDC | TSP & CVRP | 50 for Reviser-100 (with 50 initial solutions) |
| GLOP | TSP
CVRP | 100/50/10 for Reviser-100/50/20
5 for LKH3 as Reviser-20 |

### 2. A Note on Comparing Computational Budgets

We would like to respectfully clarify that directly comparing 'iteration counts' may be challenging, as the definition and computational cost of an "iteration" differ substantially across methods:

- Improvement-based methods like NeurOpt perform very fast, local search operations (e.g., k-opt) in each iteration.

- Two-stage methods like GLOP and UDC involve multiple sub-problem solvers per iteration or perform the destroy-and-repair on multiple initial solutions, and also may call a powerful heuristic solver like LKH3 internally.

- Our L2C-Insert performs local reconstruction only on a single solution using one neural model.

For this reason, the total inference time may offer a more holistic and equitable basis for comparing the computational budget. Our analysis is therefore centered on the solution quality (optimality gap) achieved within a given runtime. Our experimental results, when viewed in terms of total runtime, seem to support this perspective. For example, in Table 1 (TSP1K), our L2C-Insert (I=1000) achieves a 0.481% gap in 46.4 minutes. In contrast, a strong baseline like LEHD+RRC1000 requires 5.2 hours to achieve a 0.731% gap. This suggests that our method appears to offer a favorable trade-off between time and solution quality.

## H    Total Inference Time and Average Inference Time Per Instance

The apparent decrease in total inference time for larger instances is due to the different number of instances in the test datasets for each problem size.

Following the standard evaluation protocols established in prior NCO research [22, 43], the test sets for larger problems contain significantly fewer instances. For example, our TSP100 test set contains 10,000 instances, while the TSP10K and TSP100K sets contain only 16 instances each. The total

inference times reported in Tables 1 and 2 of our main paper reflect the time required to solve the entire dataset.

To provide a clearer picture, the table below breaks down the total inference time into the average time per instance.

Table 14: Add caption

| Dataset Size | TSP100 10000 instances | TSP1K 128 instances | TSP10K 16 instances | TSP100K 16 instances |
|---|---|---|---|---|
| L2C-Insert Total Inference Time | 8.0h | 46.4m | 16.64m | 26.07m |
| L2C-Insert Inference Time / instance | 2.9s | 21.8s | 62.4s | 97.8s |

As the table clearly shows, the average inference time per instance for L2C-Insert scales with the problem size as expected, increasing from 2.9 seconds for TSP100 to 97.8 seconds for TSP100K.

# I   Clarification on Method Positioning and Technical Contribution

Regarding the positioning of our method (construction heuristic vs. improvement heuristic) and its technical contributions, we make the following discussion to clarify and strengthen the core value of our paper.

The "insertion-based construction" is a well-established concept in the traditional field of Operations Research. The core contribution of our work is not the invention of the 'insertion' operation itself, but rather being the first learning-driven, insertion-based framework specifically designed for solution construction in the field of NCO for solving the vehicle routing problem, which is currently dominated by the 'appending-based' paradigm.

The make the technical contribution of this paper clear, we elaborate from the following three perspectives:

### 1. The Essence of Our Core Method: A Pure Insertion-based Construction Method

Our core model, L2C-Insert, is fundamentally a construction-based model in its design. Its task is to learn a policy $p$(positionnode, partial tour) to predict the best insertion position, given an incomplete solution (partial solution) and a node to be inserted. This process starts from an initial solution (e.g., one node) and builds a complete solution by sequentially adding unvisited nodes, which fully aligns with the definition of a construction-based neural model.

- **Model Architecture and Training:** As described in Sections 3.1 and 3.2 of the paper, our model architecture (especially the decoder) and the supervised learning scheme are designed to learn this single-step constructive action. The model learns to master the optimal "insertion" decision at each step by reconstructing a given optimal solution. This is fundamentally different from an improvement heuristic (like 2-opt), which requires a complete solution as input for iterative refinement.

- **Comparison with the Appending-based Paradigm:** The starting point of our work is precisely to overcome the limitations of the popular "appending-based paradigm" in existing NCO methods (as mentioned in the Introduction and Figure 1). "Appending" is a construction method, and so is "insertion." However, the latter offers greater flexibility and a larger solution space, which is the core of our exploration.

### 2. The Application of the Method: Distinguishing the Core Construction Model from the Inference Improvement Strategy

We would like to clarify that our adoption of the Insertion-based Local Reconstruction strategy during the inference stage (Section 3.3) is an advanced search strategy that uses our learned "construction model" as a core operator, rather than being the model itself.

This can be understood as a two-tiered structure:

- **Lower Level (Core Contribution):** A learned, efficient construction heuristic model (L2C-Insert) that knows how to intelligently "insert" nodes.

- **Upper Level (Application Strategy):** An improvement framework (akin to Large Neighborhood Search, LNS) that utilizes the lower-level construction model to iteratively "destroy and repair" a solution during inference to enhance its quality.

This practice of "embedding a learned construction model into an improvement framework" is one of the most recognized, state-of-the-art, and effective practices in the current NCO field (as referenced in the paper, e.g., LEHD [22], GLOP [40]). Our storyline does not confuse the two. Instead, it is:

- We propose a new, more flexible construction model (L2C-Insert).

- To validate its powerful performance, we follow the best practices in the field and use it as a potent "local reconstruction" tool during the inference stage.

### 3. Clear Technical Contributions

Based on the two points above, our technical contributions can be summarized more clearly as follows:

- **Paradigm Innovation:** In the NCO field, we are the first to explore and successfully implement a learning-driven "insertion-based" construction framework, breaking the reliance of previous research on "appending-based" construction. This is an important supplement and a rethinking of the existing NCO construction methodology.

- **Novel Model Architecture and Training Mechanism:** We designed a specific decoder structure (Section 3.1) to handle the complex "insertion" decision. It needs to simultaneously understand the intricate relationships between the node to be inserted, all unvisited nodes, and all possible insertion positions in the current partial solution. This is much more complex than an "appending" model, which only needs to focus on the last node. The corresponding supervised learning training scheme (Section 3.2) is also newly designed to fit the "insertion" task.

- **Synergistic Inference Strategy:** We propose a distance-based destruction strategy (Section 3.3) that is highly synergistic with the "insertion" paradigm. As described in the paper, this strategy can more effectively identify and break "bad edges" that are spatially close but sequentially distant, a flexibility that traditional "sequence-based destruction strategies" lack. The experiment (Table 5) also proves its effectiveness. This demonstrates the powerful synergy between our proposed construction paradigm and inference strategy.

In summary, our work is not a simple application of a known "insertion heuristic" to NCO. Our core is a novel construction model, and its application during the inference stage is an advanced and reasonable improvement strategy. We believe that this clear hierarchical division and the technical innovations at each level together constitute the solid technical contribution of this paper.

# J Analysis of Computational Complexity for Insertion-based and Appending-based heavy decoder-based NCO model

A formal analysis of computational complexity is essential for a thorough comparison. Here is a detailed comparison:

**Computational Complexity of Appending-based NCO (L2C-Append)**

The primary source of computational complexity of the Transformer-based model comes from the attention mechanism employed.

In a heavy decoder-based appending-based NCO model (e.g., LEHD, where heavy decoder means the decoder has six attention layers), the task at each construction step $t$ is to select one node from the $n - t$ unvisited nodes. The decoder computes a context query and attends to the embeddings of all $n$ initial nodes to determine the next node to append. This attention mechanism has a complexity of $O((n-t)^2 \cdot d)$, where $n$ is the total number of nodes and $d$ is the embedding dimension. This process is repeated for $n$ steps. Thus, the total complexity for constructing a single solution is $O(\frac{n^3}{3} \cdot d)$. In Big O notation, this is simplified to $O(n^3 d)$.

**Computational Complexity of Insertion-based NCO (L2C-Insert)**

Our proposed L2C-Insert framework also adopts a heavy decoder-based model. The task at each step $t$ is more complex. To make an informed decision, our decoder processes a much richer set of inputs simultaneously: the embedding of the current node to be inserted, the embeddings of all $n`t$ remaining unvisited nodes, and the embeddings of all $t`1$ possible insertion positions. The total number of input embeddings processed by the self-attention layers at every step is therefore $(1) + (n - t) + (t - 1) = n$. The complexity of a single construction step is $O(n^2 d)$. This step is repeated $n$ times to build the full solution. Consequently, the total theoretical complexity is $O(n^3 d)$.

**Discussion and Practical Implications**

Theoretically, both heavy-decoder paradigms exhibit a cubic complexity of $O(n^3 d)$. However, the insertion-based paradigm is computationally more intensive as it consistently processes $n$ nodes per step, whereas the appending-based approach processes a decreasing number of nodes $(n - t)$. This higher computational cost is a direct trade-off for the enhanced flexibility and expressive power of the insertion mechanism.

To ensure practical applicability to large-scale problems, we can employ a k-nearest neighbor (k-NN) strategy during inference. Instead of processing all $n$ contextual nodes, the decoder's attention mechanism is constrained to only the k-nearest unvisited nodes and positions. This reduces the effective sequence length from $n$ to a much smaller, constant $k$ (e.g., $k = 100$ in our paper), making the per-step complexity approximately $O(k^2 d)$.

# K    Clarification on the Definition of Target Position

We provide the following clarification to clarify the definition of the target position.

**1. Why two visited nodes in the current partial solution determine the target position**

During the process of the insertion construction, at each step, an unvisited node is inserted into a position of the partial solution constructed in the current step. Since 1) the nodes connected in the current partial solution are all visited nodes; 2) an insertion position is essentially an edge and is uniquely determined by the two nodes it connects, therefore, the only insertion position is determined by two visited points.

**2. How the target position is determined**

To make the process of identifying the target position clearer, we break it down into two steps. Given a labeled solution, a current partial solution, and a selected unvisited node to insert:

**Step 1: Identify the circular sequence in the labeled solution**

We first consider the complete labeled solution $\boldsymbol{\pi}^*$ and remove all unvisited nodes except for the currently selected one. This leaves us with a circular sequence containing only the selected unvisited node and the nodes already in the partial solution, preserving their original ordering from $\boldsymbol{\pi}^*$.

For example, suppose the labeled solution is $\boldsymbol{\pi}^* = (\pi_1, \pi_2, \pi_3, \pi_4, \pi_5, \pi_6, \pi_7)$, the partial solution is $(\pi_1, \pi_3, \pi_6, \pi_7)$, and the selected unvisited node is $\pi_2$. Then the other unvisited nodes are $\{\pi_4, \pi_5\}$. After removing $\{\pi_4, \pi_5\}$ from $\boldsymbol{\pi}^*$, the resulting circular sequence is $(\pi_1, \pi_2, \pi_3, \pi_6, \pi_7)$.

**Step 2: Identify the target position in the current partial solution**

Within this newly formed circular sequence, we can find the two nodes that are now adjacent to the selected unvisited node. These two nodes, which are part of the partial solution, define the target position for the insertion.

Continue the above example. In the circular sequence $(\pi_1, \pi_2, \pi_3, \pi_6, \pi_7)$, the nodes adjacent to $\pi_2$ are $\{\pi_1, \pi_3\}$. Since $\pi_1$ and $\pi_3$ are connected in the current partial solution, these two visited nodes determine the target position.

# L  Rationale and Principles of the Logarithmic Loss Function in Equation 3

The use of the negative log-likelihood loss function in Equation 3 is a standard and principled choice for our supervised learning framework, primarily because it is highly effective for gradient-based optimization.

Our model treats the selection of an insertion position at each construction step as a classification problem. The model's decoder, via a softmax function, outputs a probability distribution $p_\theta$ over all valid insertion positions. Our objective is to train the model to assign the highest possible probability to the correct target position $\vec{e}^*$.

The logarithmic nature of this loss function $\mathcal{L}(\boldsymbol{\theta}) = -\log p_{\boldsymbol{\theta}}\left(\vec{e}^* \mid \vec{e}_{\pi_{1:l}}, \boldsymbol{\pi}^*\right)$ provides two key advantages for training:

**Strong Error Correction:** When the model assigns a very low probability to the correct target position (i.e., $p_\theta(\vec{e}^*) \to 0$ ), the loss $-log\, p_\theta(\vec{e}^*)$ approaches infinity. This results in a very large gradient, providing a strong signal to the optimizer to correct the model's parameters significantly.

**Stable Convergence:** Conversely, as the model's prediction for the correct class approaches 1, the loss approaches 0. This leads to smaller gradients, promoting stable convergence once the model learns to make correct predictions. This behavior effectively focuses the training effort on the most difficult examples.

# M Visualization

In this section, we provide graphical examples from both TSPLIB and CVRPLIB to illustrate the insertion-based construction process clearly. As shown in Figure 5, the process begins by randomly selecting an unvisited node as the starting node for the incomplete solution. The model then continuously selects unvisited points to insert into the incomplete solution. Notably, in step 47, there are no unvisited nodes near the current node that can be connected without crossing the existing path. If the construction process is appending-based, the next step will involve directly connecting the last node to a distant unvisited node, resulting in a very long edge. However, the insertion-based construction allows for the direct insertion of the distant unvisited node into the incomplete solution (as seen in step 48), thereby avoiding the generation of the long edge mentioned above. Similarly, in Figure 6, steps 42 to 43 demonstrate a similar behavior of error correction in previous steps. This highlights the flexibility of the insertion-based construction.

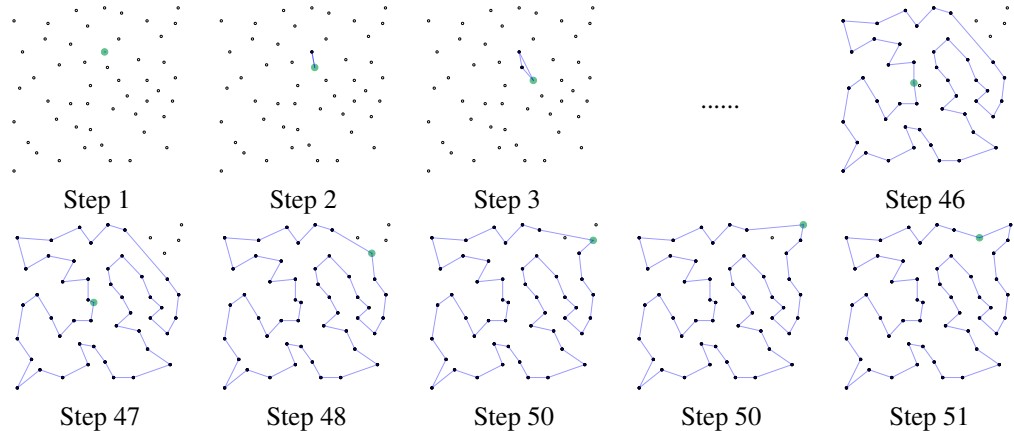

Figure 5: Insertion-based construction process on the TSPLIB instance "eil51". The green dot is the node selected in the current step for insertion. The hollow dots are the unvisited nodes. The black dots are visited nodes.

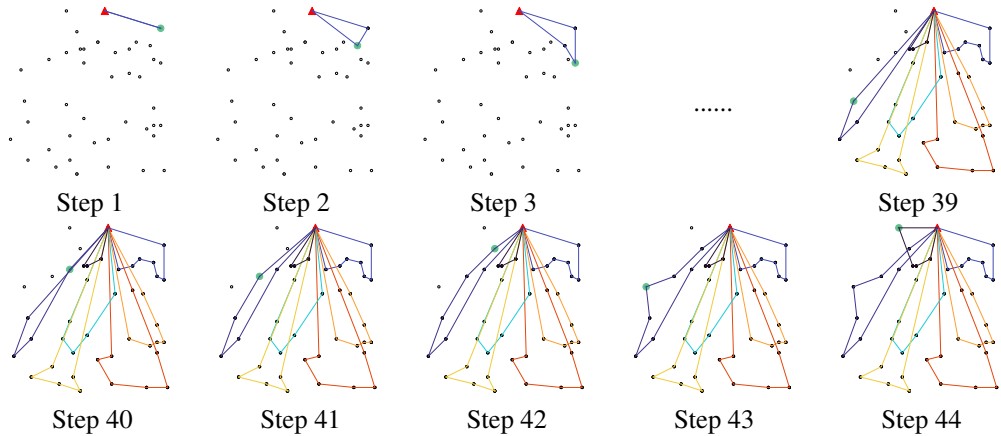

Figure 6: Insertion-based construction process on the CVRPLIB instance "A-n45-k7". The red triangle is the depot node, the green dot is the unvisited customer node selected in the current step for insertion. The hollow dots are the unvisited nodes. The black dots are visited nodes.

# N Detailed Formulation of Attention Layer

## N.1 Attention Layer

The attention layer in the transformer model [36] is widely adopted in NCO for solving VRPs, mainly including a multi-head attention (MHA) sublayer and a node-wise feed-forward (FF) sublayer. Each sublayer is followed by a skip-connection [89] and normalization (Norm) [90]. We denote $X^{(\ell-1)} \in \mathbb{R}^{n \times d}$ as the inputs of the $\ell$-th attention layer, the output of the attention layer is calculated as:

$$
\begin{aligned}
\hat{X} &= \text{Norm}(\text{MHA}(X^{(\ell-1)}) + X^{(\ell-1)}), \\
X^{(\ell)} &= \text{Norm}(\text{FF}(\hat{X}) + \hat{X}).
\end{aligned}
\tag{8}
$$

We denote this calculation process as follows:

$$
X^{(\ell)} = \text{AttnLayer}(X^{(\ell-1)}).
\tag{9}
$$

In this paper, the normalization is removed from the attention layer for enhancing generalization performance following Luo et al. [22]. For readability, we omit the $(\ell)$ and $(\ell-1)$ in the following context.

## N.2 Multi-head Attention

We first describe the single-head self-attention function as follows:

$$
\text{Attn}(X) = \text{softmax}\left(\frac{XW_Q(XW_K)^\intercal}{\sqrt{d}}\right) XW_V,
\tag{10}
$$

where $X \in \mathbb{R}^{n \times d}$ is the input matrice, and $W_Q \in \mathbb{R}^{d \times d_k}, W_K \in \mathbb{R}^{d \times d_k}, W_V \in \mathbb{R}^{d \times d_v}$ are learnable parameters.

The multi-head attention sublayer applies the single-head attention function in Equation (10) for $h$ times in parallel with independent parameters:

$$
\begin{aligned}
\text{MHA}(X) &= \text{Concat}(\text{head}_1, \dots, \text{head}_h)W^O, \\
\text{head}_i &= \text{Attn}_i(X),
\end{aligned}
\tag{11}
$$

where for each of $\text{Attn}_i(X)$, $d_k = d_v = d/h$. $W^O \in \mathbb{R}^{d \times d}$ is a learnable matrix.

## N.3 Feed Forward Layer

$$
\text{FF}(X) = \max(0, XW_{f1} + b_{f1})W_{f2} + b_{f2},
\tag{12}
$$

where $W_{f1} \in \mathbb{R}^{d \times d_{ff}}, b_{f1} \in \mathbb{R}^{d_{ff}}$ and $W_{f2} \in \mathbb{R}^{d_{ff} \times d}, b_{f2} \in \mathbb{R}^d$ are learnable parameters.

## O   Broader Impacts

This research advances the field of neural combinatorial optimization by introducing the L2C-Insert framework, which employs advanced machine learning techniques to solve Vehicle Routing Problems (VRPs). We believe this framework will provide valuable insights and inspire subsequent research focused on developing more efficient and effective neural methods for VRPs. Moreover, as a general learning-based approach for solving VRPs, the proposed L2C-Insert framework is not anticipated to have specific negative societal impacts.

## P   Licenses

The licenses for the codes and the datasets used in this work are listed in Table 15.

Table 15: List of licenses for the codes and datasets we used in this work

| Resource | Type | Link | License |
|---|---|---|---|
| LKH3 [49] | Code | http://webhotel4.ruc.dk/ keld/research/LKH-3/ | Available for academic research use |
| HGS [42] | Code | https://github.com/chkwon/PyHygese | MIT License |
| Concorde [41] | Code | https://github.com/jvkersch/pyconcorde | BSD 3-Clause License |
| POMO [21] | Code | https://github.com/yd-kwon/POMO | MIT License |
| Att-GCN+MCTS [43] | Code | https://github.com/SaneLYX/TSP_Att-GCRN-MCTS | MIT License |
| LEHD [22] | Code | https://github.com/CIAM-Group/NCO_code/tree/main/ single_objective/LEHD | MIT License |
| BQ [23] | Code | https://github.com/naver/bq-nco | CC BY-NC-SA 4.0 |
| GLOP [40] | Code | https://github.com/henry-yeh/GLOP | MIT License |
| HTSP [53] | Code | https://github.com/Learning4Optimization-HUST/H-TSP | MIT License |
| DIMES [73] | Code | https://github.com/DIMESTeam/DIMES | MIT License |
| DIFUSCO [51] | Code | https://github.com/Edward-Sun/DIFUSCO | MIT License |
| UDC [32] | Code | https://github.com/CIAM-Group/NCO_code/tree/main/ single_objective/UDC-Large-scale-CO-master | MIT License |
| ELG [30] | Code | https://github.com/gaocrr/ELG | MIT License |
| INViT [50] | Code | https://github.com/Kasumigaoka-Utaha/INViT | Available for academic research use |
| NeurOpt [55] | Code | https://github.com/yining043/NeuOpt | MIT License |
| TSPLib [46] | Dataset | http://comopt.ifi.uni-heidelberg.de/software/TSPLIB95/ | Available for any non-commercial use |
| CVRPLib [47] | Dataset | http://vrp.galgos.inf.puc-rio.br/index.php/en/ | Available for academic research use |

