# OpenReview forum: "Learning to Insert for Constructive Neural Vehicle Routing Solver"
_NeurIPS.cc/2025/Conference — NeurIPS 2025 poster_

### Official Review · Reviewer_fNy1 · 2025-06-29

**Clarity:** 3
**Significance:** 3
**Originality:** 4
**Rating:** 5
**Confidence:** 3

**Summary:**

This paper proposes L2C-Insert, the first constructive Neural Combinatorial Optimisation (NCO) framework that learns insertion to address the inherent rigidity of the prevailing append operation. Its Transformer encoder-decoder jointly embeds the current node, all candidate positions, and the remaining unvisited nodes to output a masked softmax over valid insertion positions. Supervised learning is used to train the model to learn to re-insert nodes into their ground-truth positions as defined by an optimal solution , while a distance-based destroy-reconstruct loop iteratively refines solutions at inference time. Notably, a single model trained solely on 100-node synthetic data generalizes to Travelling Salesman and Capacitated VRP instances of up to 100K nodes , achieving consistently superior performance and lower optimality gaps than recent neural and hybrid baselines

**Questions:**

Please see the above sections.

**Ethical Concerns:**

["NO or VERY MINOR ethics concerns only"]

**Final Justification:**

The authors' rebuttal successfully resolved all primary concerns, and I recommend acceptance.

**Limitations:**

Well described in the paper

**Quality:**

3

**Strengths And Weaknesses:**

# Strengths

S1. The paper introduces *insertion* as the learning target, replacing the usual tail-appending step; this is a new idea for neural combinatorial optimisation and it is good to see it explored in the field.

S2. The paper's insertion-based local reconstruction is a powerful method that is proven to improve the final solution.

S3. The paper is well organised the method is explained clearly, and the results are backed by solid evidence.

# Weaknesses
W1 . The node to be inserted is selected using a simple nearest-neighbor heuristic, not learned by the model. This greedy choice could be a limitation, and it would be interesting to know if learning this selection was explored.

W2. The paper's quality gains seem heavily dependent on the iterative destroy-reconstruct loop. On TSP100 instances, the one-shot Greedy version is notably outperformed by several NCO baselines, which makes it important to distinguish the search contribution from the policy's contribution.

W3. The runtime comparisons are difficult to fairly assess, as the search iteration budget for competing iterative methods (like NeurOpt ) is not reported. Clarifying these settings is crucial for comparing against the L2C-Insert's 1000 reconstruction steps.

---

> ### Author Rebuttal · Authors · 2025-07-31
>
> Dear Reviewer fNy1,
>
> Thank you very much for your time and effort in reviewing our work. We are glad that you consider the introduction of insertion as a learning target for NCO a novel idea worth exploring, and also find the paper to be well-organized with a clearly explained method.
>
> We address your concerns as follows.
>
> > **C1. The node to be inserted is selected using a simple nearest-neighbor heuristic, not learned by the model. This greedy choice could be a limitation, and it would be interesting to know if learning this selection was explored.**
>
>
> Thank you very much for this insightful comment. We agree that the choice of which node to insert is a critical component of the overall construction process. Our decision to use a nearest-neighbor heuristic aims to first establish the core value of the insertion paradigm itself.
>
> **1. Effectiveness of the current heuristic strategy**
>
> Our ablation study in Appendix E.2 (Table 9) provides empirical support for this choice. We compare the "Nearest Neighbor" strategy with a "Random" selection strategy and found that the nearest-neighbor approach yields substantially better performance. This indicates that this simple, efficient heuristic already provides a strong guiding signal for the construction process, making it an effective baseline.
>
> **2. Considerations for not yet adopting a learning strategy**
>
> While a learned selection policy could potentially unlock further performance gains, we consciously deferred its exploration in this foundational work for two main reasons:
>
> + **Managing Complexity:** Introducing a learned selection mechanism, whether through a separate model or by extending the current one, would significantly increase the model's complexity and computational overhead. A separate model would pose co-training challenges, while a single model performing both the selection and insertion would face a much harder learning challenge, which could degrade performance.
>
> + **Isolating the Core Contribution:** Our primary goal was to clearly isolate and demonstrate the effectiveness of the insertion paradigm as a superior alternative to the conventional appending paradigm. By using a fixed, simple heuristic for node selection, we can more confidently attribute the performance improvements to the flexibility of the insertion mechanism itself.
>
>
> We fully agree with the reviewer that learning the node selection policy is a valuable and logical next step. Developing a policy that works in synergy with the insertion model—perhaps through joint or two-stage training—is a highly promising research direction. We will add this point to the "Limitation and Future Work" section to explicitly highlight this opportunity and hope our work will inspire the NCO community to further investigate all facets of the powerful insertion-based construction paradigm.
>
>
>
> > **C2. The paper's quality gains seem heavily dependent on the iterative destroy-reconstruct loop. On TSP100 instances, the one-shot Greedy version is notably outperformed by several NCO baselines, which makes it important to distinguish the search contribution from the policy's contribution.**
>
> Thank you very much for raising this important concern. We agree that the iterative local reconstruction is a significant factor in achieving the final reported performance. We also appreciate the opportunity to clarify our perspective: we believe the key innovation of our work, the insertion-based policy, is what unlocks the true potential of this search procedure. The policy's inherent flexibility is a key enabler for the search's effectiveness.
>
> **1. The advantage of insertion-based search**
>
> As we discuss in Section 3.3, conventional appending-based models face a fundamental limitation when used with local search. When nodes are removed during the destruction phase, they can only be re-appended to the end of the sequence during reconstruction. This rigidity makes it difficult for the model to escape local optima. For instance, reconnecting two nodes that are geometrically close but far apart in the sequence—a common requirement for improving a tour—is highly improbable.
>
> Our L2C-Insert framework directly overcomes this bottleneck. By allowing a node to be re-inserted at any valid position, our model facilitates a much more powerful and effective exploration of the solution space during the reconstruction phase. This flexibility is the core reason why the destroy-and-reconstruct loop is effective in our framework.
>
> **2. Direct Ablation Study: Isolating the Policy's Contribution**
>
> To provide a direct and fair comparison that disentangles the policy's contribution, we conducted the ablation study presented in Table 4 in the main paper. In this experiment, we compared our L2C-Insert model against an L2C-Append model (which uses a similar architecture but an appending policy) within the exact same local reconstruction framework.
>
> The results show that our L2C-Insert with only 200 iterations significantly outperforms the L2C-Append model that uses 1000 iterations. This finding directly demonstrates that the superior performance stems from the intrinsic quality and flexibility of the insertion policy itself, and not just from the application of a search method.
>
>
> > **C3. The runtime comparisons are difficult to fairly assess, as the search iteration budget for competing iterative methods (like NeurOpt ) is not reported. Clarifying these settings is crucial for comparing against the L2C-Insert's 1000 reconstruction steps.**
>
> Thank you very much for your constructive suggestions. We agree that clarifying the iteration budget details of baseline methods would help to fairly evaluate the runtimes.
>
> **1. Clarifying Iteration Settings of Baselines**
>
> To clarify, we summarize the iteration settings of iterative methods in Table 4 below. From this table, we can observe that he number of iterations varies significantly. For instance, NeurOpt utilizes 10,000 lightweight k-opt iterations, while two-stage methods like UDC and SO-mixed perform 50-100 divide-and-conquer iterations.
>
>
> Table 4: Iteration settings for baselilne methods. "Reviser-$n$" refers to a solver designed to repair a local solution of length $n$.
> |Method|Problem|Number of iteration|
> |:---:|:---:|:---:|
> |NeurOpt|TSP \& CVRP|10000 (k-opt)|
> |SO-mixed|TSP1K|50 (Reviser-100/50)|
> | |TSP10K|100 (Reviser-100/50)|
> |UDC|TSP \& CVRP|50 for Reviser-100 (with 50 initial solutions)|
> |GLOP|TSP|100/50/10 for Reviser-100/50/20|
> | |CVRP|5 for LKH3 as Reviser-20|
>
>
> **2. A Note on Comparing Computational Budgets**
>
> We agree that a fair comparison is essential. In this context, we would like to respectfully clarify that directly comparing 'iteration counts' may be challenging, as the definition and computational cost of an "iteration" differ substantially across methods:
>
> + Improvement-based methods like NeurOpt perform very fast, local search operations (e.g., k-opt) in each iteration.
> + Two-stage methods like GLOP and UDC involve multiple sub-problem solvers per iteration or perform the destroy-and-repair on multiple initial solutions, and also may call a powerful heuristic solver like LKH3 internally.
> + Our L2C-Insert performs local reconstruction only on a single solution using one neural model.
>
> For this reason, the total inference time may offer a more holistic and equitable basis for comparing the computational budget. Our analysis is therefore centered on the solution quality (optimality gap) achieved within a given runtime. Our experimental results, when viewed through the lens of total runtime, seem to support this perspective. For example, from the results in Table 1 in the main paper, we can observe that on TSP1K, our L2C-Insert (I=1000) achieves a 0.481\% gap in 46.4 minutes. In contrast, a strong baseline like LEHD+RRC1000 requires 5.2 hours to achieve a 0.731\% gap. This suggests that our method appears to offer a favorable time-quality trade-off.
>
> We will carefully incorporate the above discussion into our revised paper for better clarification.
>
>
> Once again, we would like to express our heartfelt gratitude for the time and effort you've dedicated to reviewing our work. We will carefully incorporate the above discussions into our revised paper. We sincerely hope that our response can address your concerns.

---

> > ### Comment · Reviewer_fNy1 · 2025-08-06
> >
> > First, I want to thank the authors for their thorough rebuttal and the detailed ablation studies. They addressed my main concerns, and while I understand the authors’ rationale for the node‐selection heuristic (i.e., nearest neighbor), I believe the paper would benefit from a more in‐depth discussion of this aspect. In conclusion, the proposed framework is a compelling and novel idea, and I have raised my score accordingly.

---

> > > ### Author Response · Authors · 2025-08-08
> > > **Thank you very much**
> > >
> > > Dear Reviewer fNy1,
> > >
> > > Thank you very much for your time and effort in reviewing our paper and engaging with us in the discussion. We are glad to know that our responses have addressed your main concerns and you have raised the score accordingly. We will provide a more in-depth discussion of the rationale for the node-selection heuristic following your suggestion.
> > >
> > > Best Regards,
> > >
> > > Submission 27696 Authors

---

### Official Review · Reviewer_havL · 2025-06-30

**Clarity:** 3
**Significance:** 4
**Originality:** 3
**Rating:** 5
**Confidence:** 3

**Summary:**

The paper proposed an insertion-based approach to Neural Combinatorial Optimisation for solving VRP (including TSP) problems. This idea is interesting as it replaced the existing appending-based approaches, and it is a natural extension of the later considering there are existing interstation-based approaches for solving VRP problems. The model used encoder-decoder architecture, and learnt three embeddings: current node, unvisited node, and position embeddings.

**Questions:**

Why was the logarithm loss function used in Equation 3?

Target positions: it is said: “The target position for an unvisited node is defined by the two visited nodes in the current partial solution that are adjacent to this unvisited node in the complete labelled solution”   I wonder why “two visited nodes”.

Table 1:  I wonder why L2C-Insert methods took longer time on TSP100 compared to TSP1K and TSP10K?

**Ethical Concerns:**

["NO or VERY MINOR ethics concerns only"]

**Final Justification:**

This is an interesting piece of work and the idea is novel. The experimental results were solid.  This paper can be accepted by the conference.

**Limitations:**

Yes.

**Paper Formatting Concerns:**

No.

**Quality:**

4

**Strengths And Weaknesses:**

Strength:
The idea is novel and sensible, and the experimental results show the proposed model has promising potential.
The presentation for the design of the model and experiments are done in a systematic and comprehensive way.
Examples were given to better understand how insertion-based approaches work.

Weakness:
Computational complexity analysis on insertion-based against appending-based are not presented.
More detailed justification on the definition of target positions in Section 3.2.
The loss function: more details could be given.

---

> ### Author Rebuttal · Authors · 2025-07-31
>
> Dear Reviewer havL,
>
> Thank you very much for your time and effort in reviewing our work. We are glad you find that our idea of learning to insert is novel and sensible, that the proposed model has promising potential, and that the presentation of the model design and experiments is systematic and comprehensive.
>
> We address your concerns as follows.
>
> > **C1. Computational complexity analysis on insertion-based against appending-based are not presented.**
>
> Thank you for raising this crucial point. For a thorough comparison, we provide the following computational complexity analysis of insertion-based versus appending-based models:
>
> **1. Computational Complexity of Appending-based NCO (L2C-Append)**
>
> While both appending-based and insertion-based models can have the same theoretical complexity, how they arrive at it differs based on the nature of their constructive step.
>
> In a heavy decoder-based appending-based NCO model (e.g., LEHD, where heavy decoder means the decoder has six attention layers), the task at each construction step $t$ is to select one node from the $n - t$ unvisited nodes. The decoder computes a context query and attends to the embeddings of all $n$ initial nodes to determine the next node to append. This attention mechanism has a complexity of $O((n-t)^2 \cdot d)$, where $n$ is the total number of nodes and $d$ is the embedding dimension. This process is repeated for $n$ steps. Summing this over all $n$ construction steps results in a theoretical total complexity of $O(n^3⋅d)$.
>
> **2. Computational Complexity of Insertion-based NCO (L2C-Insert)**
>
> Our proposed L2C-Insert framework also adopts a heavy decoder-based model. The task at each step $t$ is more complex. To make an informed decision, our decoder processes a much richer set of inputs simultaneously: the embedding of the current node to be inserted, the embeddings of all $n − t$ remaining unvisited nodes, and the embeddings of all $t − 1$ possible insertion positions. The total number of input embeddings processed by the self-attention layers at every step is therefore $(1)+(n - t)+(t - 1)=n$. The complexity of a single construction step is $O(n^2⋅d)$. This step is repeated $n$ times to build the full solution. Consequently, the total theoretical complexity is $O(n^3⋅d)$.
>
> **3. Discussion and Practical Implications**
>
> Theoretically, both heavy-decoder paradigms exhibit a cubic complexity of $O(n^3⋅d)$. To ensure practical applicability to large-scale problems, we can employ a k-nearest neighbor (k-NN) strategy during inference, as detailed in Section 4 in the main paper. Instead of processing all $n$ contextual nodes, the decoder's attention mechanism is constrained to only the k-nearest unvisited nodes and positions. This reduces the effective sequence length from $n$ to a much smaller, constant $k$ (e.g., $k=100$ in our paper), making the per-step complexity approximately $O(k^2⋅d)$. This practical optimization makes our approach's runtime competitive with other state-of-the-art methods on larger-scale instances such as TSP10K instances, as shown in Tables 1 and 2 in the main paper.
>
> > **C2. More detailed justification on the definition of target positions in Section 3.2.**
>
> > **Q2. Target positions: it is said: “The target position for an unvisited node is defined by the two visited nodes in the current partial solution that are adjacent to this unvisited node in the complete labelled solution” I wonder why “two visited nodes”.**
>
> Thank you very much for your insightful comment. We apologize for the unclear definition of the target position. We make the following clarification to address your question.
>
> **1. Why two visited nodes in the current partial solution determine the target position**
>
> During the process of the insertion construction, at each step, an unvisited node is inserted into a position of the partial solution constructed in the current step. Since 1) the nodes connected in the current partial solution are all visited nodes; 2) an insertion position is essentially an edge and is uniquely determined by the two nodes it connects, therefore, the only insertion position is determined by two visited points.
>
> **2. How the target position is determined**
>
> To make the process of identifying the target position clearer, we break it down into two steps. Given a labeled solution, a current partial solution, and a selected unvisited node to insert:
>
> + **Step 1: Identify the circular sequence in the labeled solution**
>
>  We first consider the complete labeled solution $\boldsymbol{\pi}^\star$ and remove all unvisited nodes except for the currently selected one. This leaves us with a circular sequence containing only the selected unvisited node and the nodes already in the partial solution, preserving their original ordering from $\boldsymbol{\pi}^\star$.
>
> For example, suppose the labeled solution is $\boldsymbol{\pi}^\star=(\pi_1, \pi_2, \pi_3, \pi_4, \pi_5, \pi_6, \pi_7)$, the partial solution is $(\pi_1, \pi_3, \pi_6, \pi_7)$, and the selected unvisited node is $\pi_2$. Then the other unvisited nodes are $\{ \pi_4, \pi_5\}$. After removing $\{ \pi_4, \pi_5\}$ from $\boldsymbol{\pi}^\star$, the resulting circular sequence is $(\pi_1, \pi_2, \pi_3, \pi_6, \pi_7)$.
>
> + **Step 2: Identify the target position in the current partial solution**
>
> Within this newly formed circular sequence, we can find the two nodes that are now adjacent to the selected unvisited node. These two nodes, which are part of the partial solution, define the target position for the insertion.
>
> Continue the above example. In the circular sequence $(\pi_1, \pi_2, \pi_3, \pi_6, \pi_7)$, the nodes adjacent to $\pi_2$ are $\{\pi_1,\pi_3\}$. Since $\pi_1$ and $\pi_3$ are connected in the current partial solution, these two visited nodes determine the target position.
>
> We sincerely hope that our clarification can address this concern. We will carefully incorporate the above discussion into the revised paper to enhance the clarification.
>
> > **C3. The loss function: more details could be given.**
>
> > **Q1. Why was the logarithm loss function used in Equation 3?**
>
> Thank you very much for the valuable comment. The use of the negative log-likelihood loss function in Equation (3) is a standard and principled choice for our supervised learning framework, primarily because it is highly effective for gradient-based optimization.
>
> Our model treats the selection of an insertion position at each construction step as a classification problem. The model's decoder, via a softmax function, outputs a probability distribution $p_{\theta}$ over all valid insertion positions. Our objective is to train the model to assign the highest possible probability to the correct target position $\vec{e}^{\star}$.
>
> The logarithmic nature of the loss function provides two key advantages for training:
>
> + **Strong Error Correction:** When the model assigns a very low probability to the correct target position (i.e., $p_{\theta}(\vec{e}^{\star})\rightarrow$0 ), the loss $-log \ p_{\theta}(\vec{e}^{\star})$ approaches infinity. This results in a very large gradient, providing a strong signal to the optimizer to correct the model's parameters significantly.
>
> + **Stable Convergence:** Conversely, as the model's prediction for the correct class approaches 1, the loss approaches 0. This leads to smaller gradients, promoting stable convergence once the model learns to make correct predictions. This behavior effectively focuses the training effort on the most difficult examples.
>
> > **Q3. Table 1: I wonder why L2C-Insert methods took longer time on TSP100 compared to TSP1K and TSP10K?**
>
> Thank you very much for raising this important question.
> The apparent decrease in total inference time for larger instances is due to the different number of instances in the test datasets for each problem size.
>
> Following the standard evaluation protocols established in prior NCO research [1,2], the test sets for larger problems contain significantly fewer instances. For example, our TSP100 test set contains 10,000 instances, while the TSP1K and TSP10K sets contain only 128 and 16 instances each, respectively. The total inference times reported in Table 1 of our main paper reflect the time required to solve the entire dataset.
>
> To provide a clearer picture, table 3 breaks down the total inference time into the average time per instance.
>
> Table 3: Toal and Average Inference Time of L2C-Insert on TSP Datasets of Varying Scales
> ||TSP100|TSP1K|TSP10K|
> |-|-|-|-|
> |Dataset Size|10,000 instances|128 instances|16 instances|
> |Total Inference Time|8.0h|46.4m|16.64m|
> |Inference Time/instance|2.9s|21.8s|62.4s|
>
> As the table clearly shows, the average inference time per instance for L2C-Insert scales with the problem size as expected, increasing from 2.9 seconds for TSP100 to 62.4 seconds for TSP10K. We apologize for the lack of clarity in our initial presentation. In the revised manuscript, we will add a note to the results table to explicitly state that the reported times are for the entire test set and will consider adding the per-instance average time to avoid any future confusion. Thank you for helping us improve our paper.
>
> [1] Neural combinatorial optimization with heavy decoder: Toward large scale generalization. NeurIPS, 2023.
>
> [2] Generalize a small pre-trained model to arbitrarily large tsp instances. AAAI, 2021.
>
>
> Once again, we would like to express our heartfelt gratitude for the time and effort you've dedicated to reviewing our work. We will carefully incorporate the above discussions into our revised paper. We sincerely hope that our response can address your concerns.

---

> > ### Comment · Reviewer_havL · 2025-08-04
> > **no other concerns**
> >
> > You have responded to all my questions well, and no further questions.

---

> > > ### Author Response · Authors · 2025-08-04
> > > **Thank you very much**
> > >
> > > Thank you very much for your effort in reviewing our paper and engaging with us in the discussion. We are glad to know that our responses have addressed all your questions.

---

### Official Review · Reviewer_hZDB · 2025-07-03

**Clarity:** 2
**Significance:** 2
**Originality:** 2
**Rating:** 4
**Confidence:** 5

**Summary:**

The paper explores an insertion-based variant of Neural Combinatorial Optimization (NCO) for vehicle routing problems, proposing a method called L2C-Insert. In contrast to standard appending-based approaches that build solutions by adding nodes sequentially, the proposed method allows for inserting nodes at arbitrary feasible positions within partial solutions. The approach involves a custom model architecture for predicting insertion positions, a training procedure tailored to this setting, and an inference strategy that leverages the increased flexibility of the insertion paradigm. The authors evaluate their method on Travelling Salesman and Capacitated Vehicle Routing Problem instances, reporting improved results over existing NCO baselines. However, the extent and generality of the claimed improvements may warrant further scrutiny.

**Questions:**

Q1 Can you specify the technical contribution? Insertion-based construction is not a new concept and the way it is applied it resembles more an improvement instead of a construction method.

Q2 Can you clarify the doubts and questions listed under weaknesses with respect to the numerical experiments?

**Ethical Concerns:**

["NO or VERY MINOR ethics concerns only"]

**Final Justification:**

the authors clarified my concerns and promsied to improve the ambiguous statements. moreover they justified why they could not do further comparisons.

**Limitations:**

The paper contains a short discussion on limitations.

**Quality:**

1

**Strengths And Weaknesses:**

Strengths

S1 The topic studied is timely and general of interest


Weaknesses

W1 There are many vague and partially wrong statements in the paper. E.g., HGS is a heuristic algorithm but is claimed to provide optimal solutions.

W2 The method proposed is pitched as a construction heuristic, but is essentially used in an improvement heuristic, which misaligns the storyline with the content

W3 The results are counterintuitive and need further investigation, e.g., running an HGS on a 100 customer instance should not take 4.5 hours - with the right implementation that can be downloaded from the respective git repository, it should be possible in the order of seconds. Such kind of question marks put the overall experiments in question.

W4 Some benchmarks, e.g., the recent works of Tierney and coauthors are not considered in the numerical study although these outperform the used benchmarks.

---

> ### Author Rebuttal · Authors · 2025-07-31
>
> Dear Reviewer hZDB,
>
> Thank you very much for your time and effort in reviewing our work. We are glad to know you find the topic studied in our paper timely and generally of interest.
>
> We address your concerns as follows.
>
>
> > **C1. There are many vague and partially wrong statements in the paper. E.g., HGS is a heuristic algorithm but is claimed to provide optimal solutions.**
>
> Thank you very much for pointing this out. We agree with you that HGS is a state-of-the-art heuristic solver for CVRP, not an exact one. We apologize for the incorrect statement. In the revised manuscript, we will correct this statement. Furthermore, we will conduct a thorough review of the entire manuscript to identify and rectify any other vague or imprecise statements.
>
>
> > **C2. The method proposed is pitched as a construction heuristic, but is essentially used in an improvement heuristic, which misaligns the storyline with the content.**
>
> > **Q1. Can you specify the technical contribution? Insertion-based construction is not a new concept and the way it is applied it resembles more an improvement instead of a construction method.**
>
> Thank you very much for raising this important concern. Your questions regarding the positioning of our method (construction heuristic vs. improvement heuristic) and its technical contributions have been instrumental in helping us further clarify and strengthen the core value of our paper. We hope the following clarification will articulate our work more clearly and address your concerns.
>
> We fully agree that "insertion-based construction" is a well-established concept in the traditional field of Operations Research. The core contribution of our work is not the invention of the 'insertion' operation itself, but rather being the first learning-driven, insertion-based framework specifically designed for autoregressive solution construction in the field of NCO for solving the vehicle routing problem, which is currently dominated by the 'appending-based' paradigm.
>
> To address your concerns more clearly, we will elaborate from the following three perspectives:
>
> **1. The Essence of Our Core Method: A Pure Insertion-based Autoregressive Construction Method**
>
> Our core model, L2C-Insert, is fundamentally an autoregressive construction-based model in its design. Its task is to learn a policy $p$(position∣node, partial tour) to predict the best insertion position, given an incomplete solution (partial solution) and a node to be inserted. This process starts from an initial solution (e.g., one node) and builds a complete solution by sequentially adding unvisited nodes, which fully aligns with the definition of a construction-based neural model.
> + **Model Architecture and Training:** As described in Sections 3.1 and 3.2 of the paper, our model architecture (especially the decoder) and the supervised learning scheme are designed to learn this single-step constructive action. The model learns to master the optimal "insertion" decision at each step by reconstructing a given optimal solution. This is fundamentally different from an improvement heuristic (like 2-opt), which requires a complete solution as input for iterative refinement.
> + **Comparison with the Appending-based Paradigm:** The starting point of our work is precisely to overcome the limitations of the popular "appending-based paradigm" in existing NCO methods (as mentioned in the Introduction and Figure 1). "Appending" is a construction method, and so is "insertion." However, the latter offers greater flexibility and a larger solution space, which is the core of our exploration.
>
> **2. The Application of the Method: Distinguishing the Core Construction Model from the Inference Improvement Strategy**
>
> We would like to clarify that our adoption of the Insertion-based Local Reconstruction strategy during the inference stage (Section 3.3) is an advanced search strategy that uses our learned "construction model" as a core operator, rather than being the model itself.
>
> This can be understood as a two-tiered structure:
>
> + **Lower Level (Core Contribution):** A learned, efficient construction heuristic model (L2C-Insert) that knows how to intelligently "insert" nodes.
> + **Upper Level (Application Strategy):** An improvement framework (akin to Large Neighborhood Search, LNS) that utilizes the lower-level construction model to iteratively "destroy and repair" a solution during inference to enhance its quality.
>
> This practice of "embedding a learned construction model into an improvement framework" is one of the most recognized, state-of-the-art, and effective practices in the current NCO field (as referenced in the paper, e.g., LEHD [1], GLOP [2]). Our storyline does not confuse the two. Instead, it is:
>
> + We propose a new, more flexible construction model (L2C-Insert).
> + To validate its powerful performance, we follow the best practices in the field and use it as a potent "local reconstruction" tool during the inference stage.
>
> **3. Clear Technical Contributions**
>
> Based on the two points above, our technical contributions can be summarized more clearly as follows:
>
> + **Paradigm Innovation:** In the NCO field, we are the first to explore and successfully implement a learning-driven "insertion-based" construction framework, breaking the reliance of previous research on "appending-based" construction. This is an important supplement and a rethinking of the existing NCO construction methodology.
> + **Novel Model Architecture and Training Mechanism:** We designed a specific decoder structure (Section 3.1) to handle the complex "insertion" decision. It needs to simultaneously understand the intricate relationships between the node to be inserted, all unvisited nodes, and all possible insertion positions in the current partial solution. This is much more complex than an "appending" model, which only needs to focus on the last node. The corresponding supervised learning training scheme (Section 3.2) is also newly designed to fit the "insertion" task.
> + **Synergistic Inference Strategy:** We propose a distance-based destruction strategy (Section 3.3) that is highly synergistic with the "insertion" paradigm. As described in the paper, this strategy can more effectively identify and break "bad edges" that are spatially close but sequentially distant, a flexibility that traditional "sequence-based destruction strategies" lack. The experiment (Table 5 in the main paper) also proves its effectiveness. This demonstrates the powerful synergy between our proposed construction paradigm and inference strategy.
>
>
> In summary, our work is not a simple application of a known "insertion heuristic" to NCO. Our core is a novel construction model, and its application during the inference stage is an advanced and reasonable improvement strategy. We believe that this clear hierarchical division and the technical innovations at each level together constitute the solid technical contribution of this paper.
>
> Thank you again for your valuable comment. We hope the above explanation can address your concerns.
>
> > **C3. The results are counterintuitive and need further investigation, e.g., running an HGS on a 100 customer instance should not take 4.5 hours - with the right implementation that can be downloaded from the respective git repository, it should be possible in the order of seconds.**
>
>
> Thank you very much for raising this concern. We would like to clarify that the 4.5 hours reported in Table 1 in the main paper is the total time required for HGS to solve the entire CVRP100 test set, which consists of 10,000 instances, not the runtime for a single instance. We follow the practice of some representative literature by reporting the cumulative time over the entire test set [1, 2].
>
> If we calculate the average time, the time to solve a single CVRP100 instance is approximately 2s/instance. This average runtime is consistent with the "order of seconds" expectation mentioned by you. We sincerely apologize for this confusion. To present the results more clearly, we will explicitly state in the table captions or footnotes that the reported times for classical solvers (like HGS and LKH3) are the total times for solving the entire test set.
>
> [1] Attention, Learn to Solve Routing Problems! ICLR, 2019.
>
> [2] POMO: Policy Optimization with Multiple Optima for Reinforcement Learning. NeurIPS, 2020.
>
> > **C4. Some benchmarks, e.g., the recent works of Tierney and coauthors are not considered in the numerical study although these outperform the used benchmarks.**
>
> Thank you very much for raising this concern. We fully agree that including the important works of Tierney and coauthors [1, 2] in the numerical study is crucial for a comprehensive evaluation of our work's contributions.
>
> However, we are unable to find publicly available source code for these two papers [1, 2]. Consequently, we could not perform a fair reproduction and direct numerical comparison, which is why they are not included as baselines in our experiments. To address this and ensure our paper reflects the latest progress in the field, we commit to including a detailed introduction and discussion of these two papers in the Related Work section of our final manuscript.
>
>
> [1] PolyNet: Learning diverse solution strategies for neural combinatorial optimization. ICLR, 2025.
>
> [2] Neural Deconstruction Search for Vehicle Routing Problems. Arxiv, 2025.
>
>
> Once again, we would like to express our heartfelt gratitude for the time and effort you've dedicated to reviewing our work. We will carefully incorporate the above discussions into our revised paper. We sincerely hope that our response can address your concerns.

---

> > ### Comment · Reviewer_hZDB · 2025-08-04
> >
> > I thank the authors for clarifying my concerns. I just checked the Polynet repository and saw that it is indeed empty such that the authors cannot do further comparisons. I will raise my score accordingly, assuming that all clarifications will be reflected in teh camera ready version

---

> > > ### Author Response · Authors · 2025-08-04
> > > **Thank you very much**
> > >
> > > Thank you very much for your effort in reviewing our paper and engaging with us in the discussion. We are glad to know that our responses have clarified your concerns and you have raised the score. We will ensure that all clarifications are carefully incorporated into the revised paper.

---

### Official Review · Reviewer_s3Av · 2025-07-05

**Clarity:** 3
**Significance:** 1
**Originality:** 2
**Rating:** 2
**Confidence:** 4

**Summary:**

The paper introduces an insertion heuristic based on attention neural networks for solving large vehicle routing problems. It evaluates it on large problem instances and compares it to existing heuristics.

**Questions:**

- Why does L2C-Insert gets faster as the instance size increase?

**Ethical Concerns:**

["NO or VERY MINOR ethics concerns only"]

**Limitations:**

See weaknesses.

**Paper Formatting Concerns:**

/

**Quality:**

2

**Strengths And Weaknesses:**

The paper is well-written and the approach is interesting. Extending NCOs with insertion capabilities is a valuable contribution. However, I find that the proposed experimental results do not allow for a fair assessment of the value of the proposed method. This is for a few reasons:
- the methods have a different computational budget, which vary with the instance size. However, only a single point in time is shown. The authors claims that L2C-Insert can reach speed-up of up to 8x compared to the oracle heuristic benchmarks. However, the value achieved by these heuristics with the same time budget as L2C insert is never shown.
- it is stated that L2C uses an "accelerate[d] inference by processing only the k-nearest unvisited nodes and positions of the current node". This is a valuable strategy. However, it is unclear why it can be applied only to L2C and not to all other benchmarks. This will lead to significant speed-ups on large instances and avoid OOM.
The paper is essentially an experimental one. Yet, the current experiments do not allow to fairly assess the value of the new method compared neither to existing NCOs nor HGS.

HGS is the state-of-the art heuristic. It uses a variety of moves for constructing its solutions. As shown by the authors, most NCOs focus on appending nodes to partial solutions. L2C-Insert focuses on inserting nodes. Since solvers are modular, it would be very interesting to design a NCO that combines existing appending moves with L2C-Insert.

---

> ### Author Rebuttal · Authors · 2025-07-31
>
> Dear Reviwer s3Av,
>
> Thank you very much for your time and effort in reviewing our work. We are glad to know you find our paper well-written and the approach interesting, and that extending NCOs with insertion capabilities is a valuable contribution.
>
> We address your concerns as follows.
>
> > **C1. the methods have a different computational budget, which vary with the instance size. However, only a single point in time is shown. The authors claims that L2C-Insert can reach speed-up of up to 8x compared to the oracle heuristic benchmarks. However, the value achieved by these heuristics with the same time budget as L2C insert is never shown.**
>
> Thank you very much for raising this concern. We agree that a direct, time-normalized comparison against all baselines is important for a comprehensive evaluation. We would like to make the following clarifications to address your concern.
>
> **1. On the Evaluation Protocol and Role of Heuristic Solvers**
>
> We would like to clarify the role of classical heuristic solvers (e.g., LKH3, HGS) in our experimental setup. In line with the established practice in the NCO literature, these solvers are primarily included to provide a strong, near-optimal baseline for solution quality, rather than to serve as direct competitors in terms of runtime. This evaluation protocol follows the precedent set by numerous recent publications in top-tier conferences. For example, prominent works such as POMO [1] and BQ-NCO [2] adopt a similar evaluation strategy.
>
> **2. On the Speed-up Claim**
>
> Additionally, we would like to clarify that all speed-up claims in our paper are made in comparison to other state-of-the-art Neural Combinatorial Optimization (NCO) methods, not classical heuristic solvers like LKH3 or HGS.
>
> We recognize the potential for confusion. In the revised version of our paper, we will explicitly clarify the distinct evaluation roles of classical solvers versus NCO methods in our experimental setup section.
>
> [1] POMO: Policy Optimization with Multiple Optima for Reinforcement Learning. NeurIPS, 2020.
>
> [2] BQ-NCO: Bisimulation Quotienting for Efficient Neural Combinatorial Optimization. NeurIPS, 2023.
>
>
> > **C2. it is stated that L2C uses an "accelerate[d] inference by processing only the k-nearest unvisited nodes and positions of the current node". This is a valuable strategy. However, it is unclear why it can be applied only to L2C and not to all other benchmarks. This will lead to significant speed-ups on large instances and avoid OOM.**
>
> Thank you very much for raising this concern. We agree that the acceleration techniques like k-nearest neighbors (k-NN) strategy are powerful to be applied to other constructive NCO methods. We would like to make the following clarifications to address your concern.
>
> **1. Impact of the acceleration techniques**
>
> It is challenging to uniformly apply a specific set of acceleration techniques across all baseline methods and expect consistent improvements. To provide a concrete example, we applied our acceleration settings (specifically, using k=100 nearest neighbors and limiting the destruction size to 300 for local reconstruction) to a strong baseline, LEHD RRC1000. The results are presented in Table 1 below.
>
> Table 1: Performance of LEHD RRC1000 with and without the acceleration techniques.
> ||TSP1K|||TSP10K|||TSP100K|||
> |-|:-:|:-:|:-:|:-:|:-:|:-:|:-:|:-:|:-:|
> ||Length|Gap|Time|Length|Gap|Time|Length|Gap|Time|
> |LEHD RRC1000|23.288|0.731\%|5.2h|80.900|12.709\%|3.6h|OOM|||
> |LEHD RRC1000 with acceleration techniques|23.389|1.168\%|17.62m|79.734|11.084\%|9.49m|368.023|62.849\%|15.47m|
>
> These results lead to several key observations:
>
>  + **The strategy is not a universal enhancement:** On TSP1K, applying acceleration techniques drastically reduces inference time but leads to a noticeable degradation in solution quality (Gap increases from 0.731\% to 1.168\%). This demonstrates a non-trivial trade-off between speed and performance.
> + **They enable large-scale inference but with limitations:** On TSP100K, the acceleration techniques successfully avoid the OOM error, allowing LEHD to produce a solution. However, the resulting quality (62.8\% gap) is not competitive, suggesting that simply restricting the search space is insufficient for achieving good performance on very large instances with this model.
>
> These results show that while acceleration techniques can be integrated into other methods, they are not a "free lunch". Their impact on performance is unpredictable and can even be detrimental, indicating that such strategies need to be co-designed with the model architecture.
>
>
> **2. Fair Comparison and Our Experimental Choices**
>
> Given the above, we chose to run baselines using their original, published configurations for two main reasons:
>
> + **To ensure a transparent and faithful comparison:** Modifying baselines with a strategy that has complex trade-offs could be seen as unfairly altering their intended design.
> + **Some baselines already incorporate similar acceleration techniques:** For instance, INViT-3V already incorporates a  k-NN neighborhood-based attention mechanism.
>
> In short, while we acknowledge the general applicability of acceleration techniques like k-NN, our analysis shows they are not a simple, plug-and-play solution that guarantees benefits for all models. Therefore, to maintain a fair comparison, we benchmark against the original design of the baseline methods.
>
> > **Suggestion. It would be very interesting to design a NCO that combines existing appending moves with L2C-Insert.**
>
> Thank you very much for this insightful and forward-looking suggestion. We completely agree that designing a hybrid NCO solver that combines both appending and inserting moves is a highly promising research direction. A modular framework that can dynamically learn to select the most effective operator—be it appending to the end of a tour or inserting into an optimal position—could leverage the strengths of both paradigms. Such a model has the potential to be more flexible and powerful than a model restricted to a single type of move.
>
> Nonetheless, the primary objective of this paper is to serve as an in-depth investigation into the insertion-based paradigm within the constructive NCO framework. To rigorously and clearly establish its standalone value, we intentionally isolated it and benchmarked it against the traditional, purely appending-based paradigm (as shown in our ablation study in Table 4 in the main paper, L2C-Append vs. L2C-Insert). Introducing a hybrid model from the outset would have made it difficult to disentangle the benefits and attribute performance gains directly to the insertion capability itself versus other potential synergistic effects. By focusing on a pure insertion model, we aim to provide a clean and interpretable analysis of its unique contributions.
>
> We believe your suggestion represents a natural and exciting next step for this line of research. We will explicitly add this idea to our "Limitation and Future Work" section in the revised manuscript to highlight its potential and inspire future work in this area.
>
>
> > **Q1. Why does L2C-Insert gets faster as the instance size increase?**
>
> Thank you very much for raising this important question. The apparent decrease in total inference time for larger instances is due to the different number of instances in the test datasets for each problem size.
>
> Following the standard evaluation protocols established in prior NCO research [1,2], the test sets for larger problems contain significantly fewer instances. For example, our TSP100 test set contains 10,000 instances, while the TSP10K and TSP100K sets contain only 16 instances each. The total inference times reported in Tables 1 and 2 of our main paper reflect the time required to solve the entire dataset.
>
> To provide a clearer picture, Table 2 breaks down the total inference time into the average time per instance.
>
> Table 2: Total and Average Inference Time of L2C-Insert on TSP Datasets of Varying Scales
> ||TSP100|TSP1K|TSP10K|TSP100K|
> |-|-|-|-|-|
> |Dataset Size|10,000 instances|128 instances|16 instances|16 instances|
> |Total Inference Time|8.0h|46.4m|16.64m|26.07m|
> |Inference Time/instance|2.9s|21.8s|62.4s|97.8s|
>
> As the table clearly shows, the average inference time per instance for L2C-Insert scales with the problem size as expected, increasing from 2.9 seconds for TSP100 to 97.8 seconds for TSP100K. We apologize for the lack of clarity in our initial presentation. In the revised manuscript, we will add a note to the results table to explicitly state that the reported times are for the entire test set and will consider adding the per-instance average time to avoid any future confusion.
>
>
> [1] Neural combinatorial optimization with heavy decoder: Toward large scale generalization. NeurIPS, 2023.
>
> [2] Generalize a small pre-trained model to arbitrarily large tsp instances. AAAI, 2021.
>
> Once again, we would like to express our heartfelt gratitude for the time and effort you've dedicated to reviewing our work. We will carefully incorporate the above discussions into our revised paper. We sincerely hope that our response can address your concerns.

---

### Decision · Program_Chairs · 2025-09-17

**Decision:**

Accept (poster)

**Comment:**

This paper proposes a novel insertion-based paradigm for Neural Combinatorial Optimization (NCO), as an alternative to the conventional appending-based approaches, and demonstrates promising results on the Vehicle Routing Problem. By integrating model design, training strategy, and inference algorithms, the paper consistently outperforms existing baselines, and its novelty and usefulness are evident.

The review situation is as follows:

Reviewer havL and Reviewer fNy1 praised the novelty of the idea and the persuasiveness of the experiments, ultimately recommending Accept (5).

Reviewer hZDB initially raised concerns but found them sufficiently addressed through the rebuttal, raising the score to 4 (Borderline Accept).

In contrast, Reviewer s3Av provided some technical comments but did not engage at all with the authors’ detailed rebuttal. This lack of dialogue was flagged by the Area Chair as an insufficient review. Therefore, the reliability of this review is low and should not weigh heavily in the final decision.

Overall, the paper establishes a novel paradigm with clear originality and demonstrates promising results through solid experiments. The concerns raised have been adequately addressed by the authors, and remaining weaknesses are minor and can be improved in future work.

In conclusion, I recommend this paper for acceptance at NeurIPS.